# Microglia and macrophages alterations in the CNS during acute SIV infection: A single-cell analysis in rhesus macaques

Xiaoke Xu[1], Meng Niu[1], Benjamin G. Lamberty[1], Katy Emanuel[1], Shawn Ramachandran[1], Andrew J. Trease[1], Mehnaz Tabassum[2], Jeffrey D. Lifson[3], Howard S. Fox[1]*

1 Department of Neurological Sciences, University of Nebraska Medical Center, Omaha, Nebraska, United States of America, 2 Department of Pathology, Microbiology, and Immunology, University of Nebraska Medical Center, Omaha, Nebraska, United States of America, 3 AIDS and Cancer Virus Program, Frederick National Laboratory, Frederick, Maryland, United States of America

* hfox@unmc.edu

**Data Availability Statement:** The data that support the findings of this study are publicly available from the National Center for Biotechnology Information

## Abstract

Human Immunodeficiency Virus (HIV) is widely acknowledged for its profound impact on the immune system. Although HIV primarily affects peripheral CD4 T cells, its influence on the central nervous system (CNS) cannot be overlooked. Within the brain, microglia and CNS-associated macrophages (CAMs) serve as the primary targets for HIV and the simian immunodeficiency virus (SIV) in nonhuman primates. This infection can lead to neurological effects and establish a viral reservoir. Given the gaps in our understanding of how these cells respond *in vivo* to acute CNS infection, we conducted single-cell RNA sequencing (scRNA-seq) on myeloid cells from the brains of three rhesus macaques 12 days after SIV infection, along with three uninfected controls. Our analysis revealed six distinct microglial clusters including homeostatic microglia, preactivated microglia, and activated microglia expressing high levels of inflammatory and disease-related molecules. In response to acute SIV infection, the homeostatic and preactivated microglia population decreased, while the activated and disease-related microglia increased. All microglial clusters exhibited upregulation of MHC class I molecules and interferon-related genes, indicating their crucial roles in defending against SIV during the acute phase. All microglia clusters also upregulated genes linked to cellular senescence. Additionally, we identified two distinct CAM populations: CD14$^{low}$CD16$^{hi}$ and CD14$^{hi}$CD16$^{low}$ CAMs. Interestingly, during acute SIV infection, the dominant CAM population changed to one with an inflammatory phenotype. Specific upregulated genes within one microglia and one macrophage cluster were associated with neurodegenerative pathways, suggesting potential links to neurocognitive disorders. This research sheds light on the intricate interactions between viral infection, innate immune responses, and the CNS, providing valuable insights for future investigations.

## Author summary

HIV's entry into the central nervous system (CNS) can lead to neurological dysfunction, including HIV-associated neurocognitive disorders (HAND) and the establishment of a

(NCBI) with the identifiers PP236443 (SIV genomic sequence in GenBank) and GSE253835 (single cell transcriptome sequences in GEO).

**Funding:** This work was supported by National Institutes of Health (NIH, www.nih.gov) grants (U01DA053624 to HF, R21MH128057 to HF, and P30MH062261 to HF). HF, KE, and BL received salary from U01DA053624, R21MH128057, and P30MH062261. XX, AT, and MT received salary from U01DA053624. MN received salary from U01DA053624 and R21MH128057. The funders had no role in study design, data collection and analysis, decision to publish, or preparation of the manuscript.

**Competing interests:** The authors have declared that no competing interests exist.

viral reservoir. While microglia and CNS-associated macrophages (CAMs) are the primary targets of HIV in the CNS, their responses during acute HIV infection remain poorly defined. To address this, we employed the scRNA-seq technique to study microglial and CAM populations in rhesus macaques during acute SIV infection. By identifying signature genes associated with different phenotypes and mapping them to various biological and pathological pathways, we discovered two myeloid cell clusters strongly linked to neurodegenerative disorders. Other clusters were associated with inflammatory pathways, suggesting varying degrees of activation among different myeloid cell populations in the brain, possibly mediated by distinct signaling pathways. All microglia clusters developed signs of the cellular senescence pathway. These findings shed light on the immunological and pathological effects of different myeloid phenotypes in the brain during acute SIV infection, providing valuable insights for future therapeutic strategies targeting this critical stage and aiming to eliminate the viral reservoir.

## Introduction

The human immunodeficiency virus (HIV) is an enveloped retrovirus that contains two copies of a single-stranded RNA genome, which can cause acquired immunodeficiency syndrome (AIDS) by significantly impairing the immune system. HIV remains a global health challenge with profound implications for individuals, communities, and societies. The estimated number of people with HIV (PWH) is 39.0 million (33.1–45.7 million) worldwide at the end of 2022, and two-thirds of whom (25.6 million) are in the African region [1]. In the United States, the annual number of new diagnoses decreased by 7% from 2017 to 2021, but there are still an estimated 1.2 million people who had HIV at the end of 2021 [2]. Like HIV in genomic, structural, and virologic perspectives, the simian immunodeficiency virus (SIV) also belongs to the primate retrovirus family. *In vivo*, both viruses cause persistent infection. Infection of nonhuman primates (NHPs) by SIV mimics many key aspects of HIV infection in humans, including immunodeficiency, opportunistic infections, and CNS infection, which can be associated with neurological impairments. [3–6]

The development of HIV infection can be classified into three stages: acute HIV infection, chronic HIV infection, and, if untreated, eventually AIDS [7]. The acute infection period is defined as the stage immediately after HIV infection and before the development of antibodies to HIV, which generally happens within 2 to 4 weeks after HIV infection [8,9]. For rhesus monkeys infected with $SIV_{mac}$, high-level plasma antigenemia could be detected from approximately day 7 through day 21, and the anti-$SIV_{mac}$ antibodies could be first detected in the blood at approximately day 14, which is close to the acute infection phase in human [10]. Sexually-mediated HIV transmission generally starts with mucosal CD4+ T cells and Langerhans cells [11] and then travels to gut-associated lymphoid tissue (GALT). Intravenous infection-mediated HIV transmission leads to initial infection of CD4+ T cells in lymph nodes, the spleen, and GALT [12,13]. Lymphoid tissue and other organ macrophages can also be infected. HIV-infected cells can produce large amounts of viruses to infect additional target cells and migrate and carry the virus to other tissues and organs, including the central nervous system (CNS). The earliest post-infection time for detecting HIV/SIV RNA in the CNS (brain or cerebrospinal fluid) ranges from 4 to 10 days [14–16]. Like the deteriorative effect of HIV in the periphery, the inflammatory events and neurotoxicity elicited by the HIV/SIV can damage neurons as well as other supportive cells in the brain, which can eventually lead to HIV-associated neurocognitive disorders (HAND). While the extensive use of antiretroviral therapy

(ART) has significantly reduced the occurrence of dementia, the most severe type of HAND, the overall prevalence of HAND still hovers around 50% [17–20].

In CNS HIV/SIV infection, CNS-associated macrophages (CAMs) and microglia are thought to play a central role in defending against the invading pathogen and triggering neuroinflammation [21,22]. They are activated by responding to the virus and/or virally infected cells that enter the brain as the initial innate immune response. They can release numerous proinflammatory cytokines, including interferons, IL-6, IL-1β, and TNF-α, to control and clear the virus or infected cells from the CNS [14]. However, macrophages and microglia can also contribute to the pathological events of HIV/SIV infection. In acute SIV/HIV infection, infected blood monocytes represent another cell type other than CD4+ T cell that has been proposed to seed the virus in the CNS. Infection of rhesus monkeys with SIV indeed results in an increase in monocyte trafficking to the brain [23]. Once trafficking monocytes enter the brain, they can further differentiate to CAMs, and under experimental conditions (such as depletion of CD8+ cells), can lead to the rapid development of SIV encephalitis [24]. On the other hand, blocking an integrin (α4), which is highly expressed in monocytes, by natalizumab hindered CNS infection and ameliorated neuronal injury [25]. Integrin α4 is also expressed on CD4+ T cells; thus, attributing the effect to monocytes is uncertain. Although microglia, the CNS-specific resident myeloid cells, do not bring HIV/SIV to the CNS, they are actively involved in neuronal damage once they are infected and/or activated [17,26,27]. In addition, infected CAMs and/or microglia are thought to make up a viral reservoir in the brain under suppressive ART treatment, complicating efforts for an HIV cure.

Although the general responses of microglia and CAMs to acute SIV infection have been widely studied using bulk assays, our understanding of different microglial or CAM phenotypes is minimal for this important period in which the virus enters the brain. In this study, we performed high-throughput single-cell RNA sequencing (scRNA-seq) on microglia and CAMs from the brains of rhesus macaques during acute SIV infection and in control uninfected animals to address the limitations of bulk assays. We investigated the different effects of acute SIV infection on varied myeloid phenotypes. We identified homeostatic microglia, preactivated microglia, activated microglia, and two phenotypes of CAMs in the brains. We further characterized these subsets of cells by comparing their transcriptomic profiles in the uninfected and acute-infected conditions. Different responses were evoked in acute SIV infection microglial and macrophage phenotypes. Interestingly, two activated cell clusters were closely associated with neurological disorders. Finally, although we did not observe modulation of the expression of the anti-apoptotic molecule BCL-2 (upregulation of BCL-2 is one of the mechanisms for promoting the survival of infected cells) [28,29] by SIV in microglia and CAMs, we found another anti-apoptotic molecule, CD5L was highly expressed in infected microglia which might be a novel potential pathway to elucidate the reservoir establishment in the CNS.

## Results

### The constitution of the brain's myeloid cell population was altered by acute SIV infection

To examine the effect of acute SIV infection on brain myeloid cells, we inoculated three rhesus macaques with $SIV_{mac}251$. We processed the brains at 12 days post-inoculation to enrich resident and infiltrating immune cells for scRNA-seq, given that a 12-day infection could generate a high viremia and before the anti-$SIV_{mac}$ antibodies could be detected in the blood [10]. Cells from three uninfected animals were similarly analyzed. In the infected animals, high viral loads were found in the plasma at this stage, and productive brain infection (high viral RNA/ DNA ratios) was found (S1 Fig and S1 Table). Uniform Manifold Approximation and

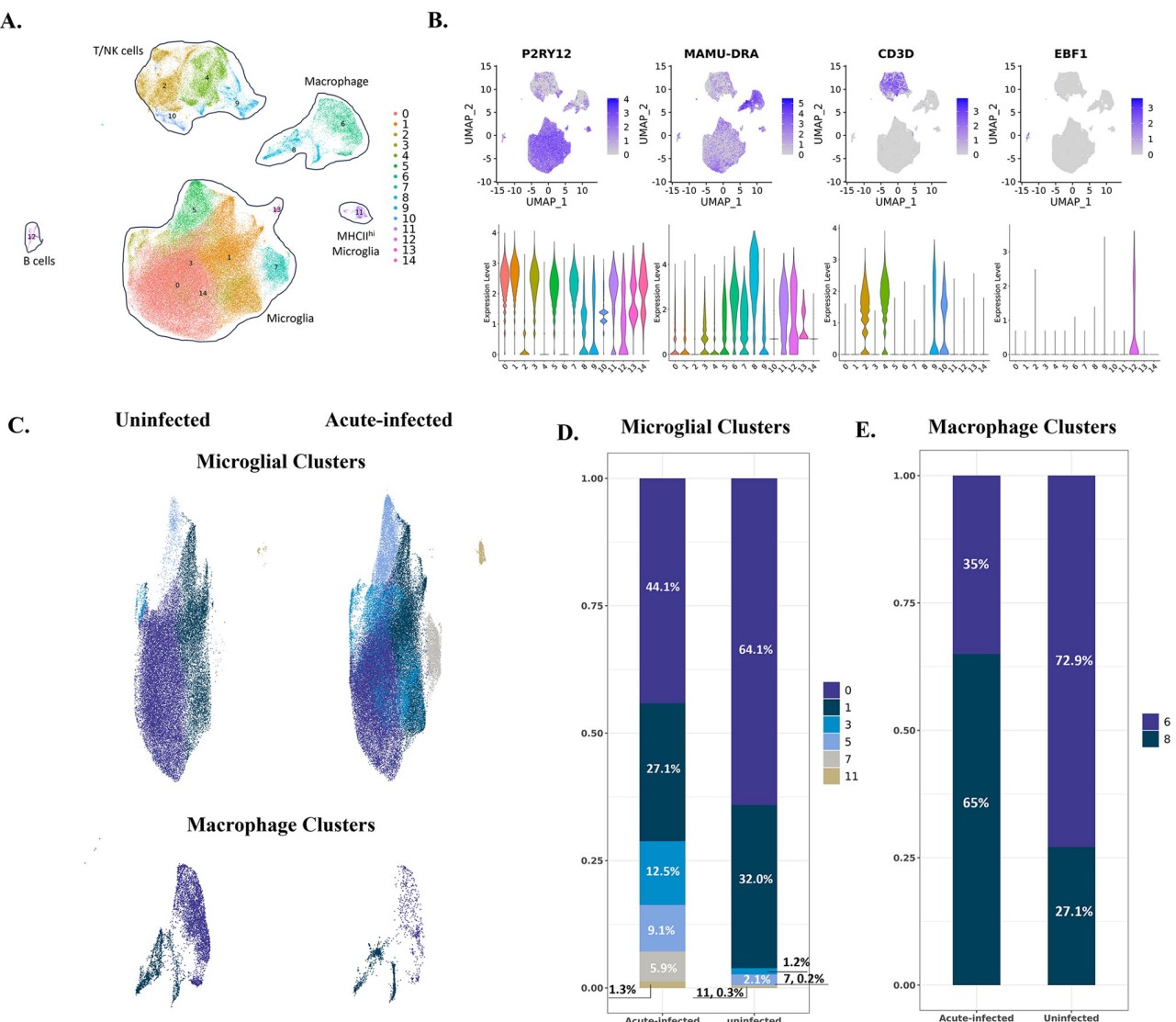

**Fig 1. The identified cell clusters in the brains of uninfected and acutely infected rhesus macaques and the population changes of myeloid cells.**
**(A)** 15 cell clusters were identified by graph-based clustering and projected in UMAP. Clusters 0, 1, 3, 5, 7, 11, 13, and 14 were microglia; Clusters 2, 4, 9, and 10 were T/NK cells; Clusters 6 and 8 were macrophages, and Cluster 12 was B cells. **(B)** The expression of representative cell markers was used to identify microglia (P2RY12), macrophages (MAMU-DRA), T cells (CD3D), and B cells (EBF1). **(C)** The comparison of myeloid cells' compositions in the brain between uninfected animals and animals with acute SIV infection. The upper UMAPs include the microglial clusters; the lower UMAPs include macrophage clusters. **(D)** Percentage of each microglial cluster in animals with acute infection and uninfected animals. **(E)** Percentage of each macrophage cluster in animals with acute infection and uninfected animals.

Projection (UMAP) visualization of the scRNA-seq data revealed that the transcriptional patterns of the cells purified from the brains of the six rhesus macaques (three uninfected and three acute SIV-infected) grouped in five distinct regions. Graph-based clustering revealed that two of those regions contained distinct clusters, whereas in the other three existed additional subclusters of cells, totaling fifteen separate clusters (Fig 1A).

After comparing the expression of different cell markers between those 15 cell clusters (Fig 1B and S2 Table), we identified eight microglial clusters with high expression of microglial core genes (P2RY12, GPR34, and CX3CR1) within the main microglia cluster and one separate

cluster of microglia-like cells that also expressed high levels of major histocompatibility complex class II (MHC II) molecules. We also found two macrophage clusters with high MHC II and S100 molecule expression (MAMU-DRA, CD74, S100A6, and S100A4). In addition, we identified four T/NK cell clusters with high expression of the genes for T cell co-receptors, granzymes, and NK cell granule proteins (CD3D, GZMB, and NKG7), and one B cell cluster with high expression of B cell markers (EBF1 and MS4A1). The scope of this study is to investigate the myeloid cells in response to acute SIV infection, so we only included the myeloid cells, microglia and CNS-associated macrophages (CAMs) for subsequent analyses.

We first assessed whether we could detect cells expressing SIV transcripts. Indeed, in both the microglia and CAM clusters, SIV-infected cells could be identified, in total, 0.15% of these myeloid cells. For the lymphocyte population in the brain, the infection rate was ~1.4% (10-fold higher than the myeloid population), which is consistent with the fact that lymphoid cells, especially CD4+ T cells, are primary leukocytes to be infected during acute SIV/HIV infection. (S1 Table) However it is difficult to assess which cells were responsible for infecting the brain, given the large number of endogenous myeloid cells present in the brain. Given the imperfect sensitivity of scRNA-seq, these percentages of infected cells may be artificially low due to false negatives. In the five regions of the brain analyzed for SIV DNA from the three acutely infected monkeys, we found an average of 162 copies of the SIV proviral genome per million cells (S1 Fig and S1 Table). If one approximates brain macrophages and microglia as comprising 10% of CNS cells, and not including the lymphocytes as the myeloid cells far outnumber the lymphoid cell in the brain, this would imply that 0.16% of these cells are infected, similar to our finding of 0.15% of these cells with SIV transcripts. Thus, at least during the acute infection stage, it is likely that the level of expression from the viral genome is sufficient to be recognized in scRNA-seq experiments. Given the similarity in the proportion of cells with SIV DNA and SIV transcripts, it is likely that most infected cells are expressing SIV transcripts, consistent with the high SIV RNA to DNA ratio (S1 Fig and S1 Table).

We next examined whether the infected myeloid cells exhibited differences in gene expression patterns. Other than SIV itself, only two genes (LOC100426197, a class I histocompatibility gene, and LOC693820, the 40S ribosomal protein S29) reached statistical significance for differential expression between the infected myeloid cells and uninfected ones in the acutely infected animals (S3 Table). This is likely due to the activation of even the uninfected cells in the infected animals, as comparing the SIV-positive cells to all the myeloid cells in the uninfected animals yielded 226 DEGs (S3 Table). Confirming the effect of the acute infection on uninfected cells, when comparing these bystander uninfected cells from the acutely infected animals to those in the uninfected animals, 428 DEGs were identified (S3 Table). 182 DEGs were in common between the last two comparisons (S2 Fig).

To better assess how acute infection alters the myeloid populations, we separately compared the microglia and macrophage clusters between the acutely infected animals and the control uninfected animals. Cluster 13 and Cluster 14 were excluded for their low cell population (Cluster 13 has 273 cells, and Cluster 14 has 177 cells) and fewer identified markers other than microglial core genes (S2 Table). Therefore, we did downstream analyses on the six microglial clusters (0: Micro-0, 1: Micro-1, 3: Micro-3, 5: Micro-5, 7: Micro-7, 11: Micro-11) and two macrophage clusters (6: Macro-6 and 8: Macro-8).

As described briefly above, the initial identification of cell phenotypes revealed that Micro-11 had a high expression of microglial core genes. Still, from the UMAP (Fig 1A), it did not aggregate with other microglial clusters. In addition, the cells in Micro-11 also had high expression of MHC class II molecules (Fig 2A). Even though MHC class II molecules typically are not highly expressed in the microglia cells compared with CAMs, [21] some microglial

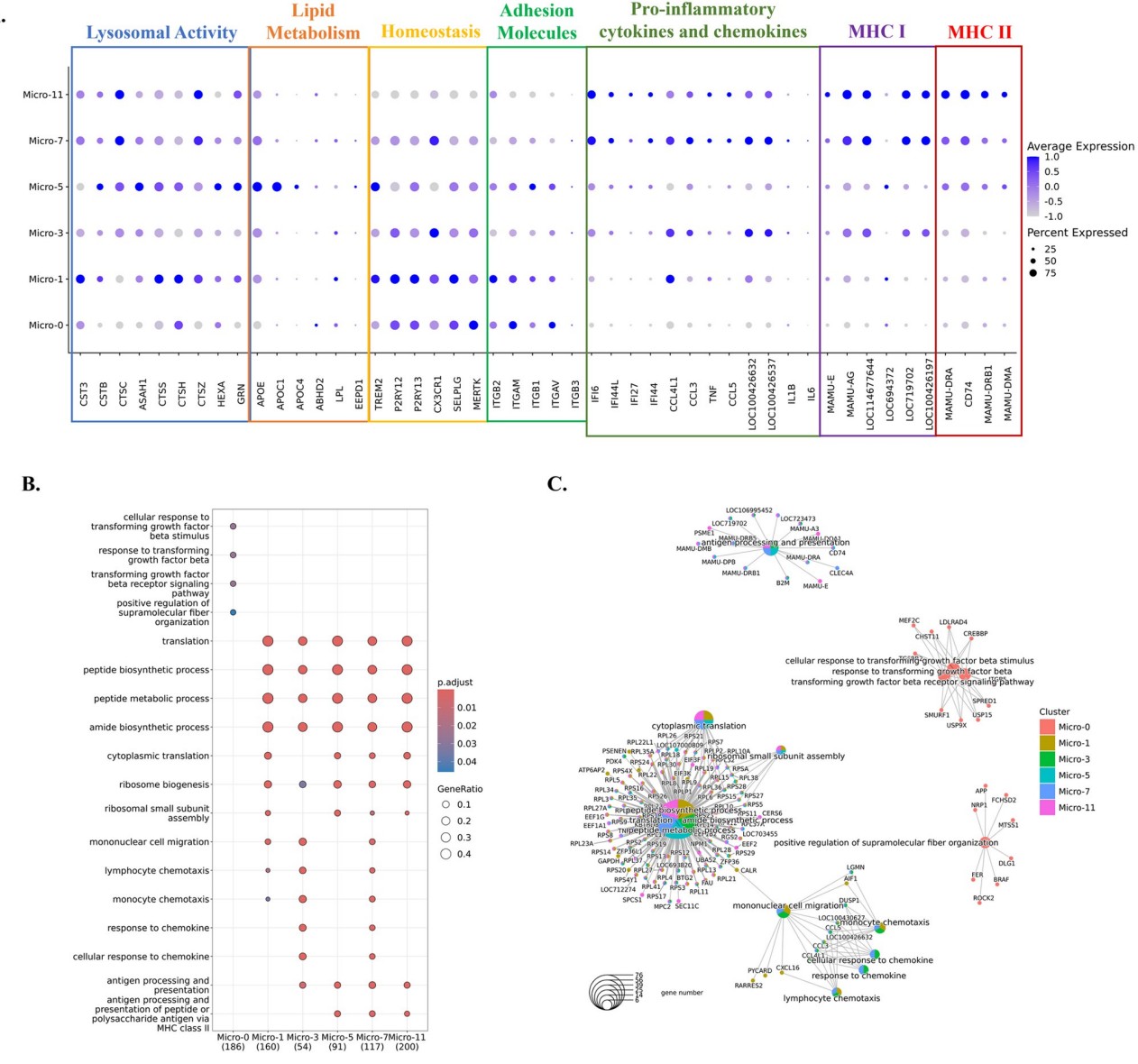

**Fig 2. Characterizations of the identified microglial clusters. (A)** Expression of selected differentially expressed genes (DEGs) in six microglial clusters. The DEGs of each microglial cluster were found using the Wilcoxon rank sum test for comparison. The genes shown were categorized by the functions related to lysosomal activity, lipid metabolism, homeostasis, adhesion molecules, proinflammatory cytokines and chemokines, MHC class I molecules, and MHC class II molecules. The data was scaled for plotting. LOC100426537: C-C motif chemokine 3-like. LOC100426632: C-C motif chemokine 4. **(B)** The characteristic biological processes in each microglial cluster by Gene Ontology (GO) pathway analysis. **(C)** The networks between DEGs and the biological processes in each microglial cluster.

cells in white matter or under pathological conditions might have higher expression [30–32]. Thus, we included this cluster in the analyses for microglia instead of CAM.

Comparing the microglial constituents between the uninfected and infected animals (Fig 1C and 1D), we found that Micro-0 and Micro-1 decreased their proportions under acute SIV infection, compensated by the increased microglial cells from the other four clusters. Furthermore, this increase in acute infection was more remarkable for Micro-3 and Micro-7, each increasing at least 10-fold, suggesting those two clusters were highly reactive to the infection.

For the macrophage clusters, Macro-6 was the predominant macrophage population compared to Macro-8 in the brains of uninfected animals (1.9-fold higher). However, this pattern was switched during SIV infection (Fig 1C and 1E). There was a 3.5-fold higher population of Macro-8 over Macro-6 in animals with acute SIV infection, suggesting the important roles of Macro-8 in reacting to early viral invasion.

## The increased microglial populations in acute SIV infection were activated microglia but they have different activation patterns and pathways

We further characterized the clusters to understand better the factors behind the changes in microglial populations during acute SIV infection. We initially compared them regarding the expression of lysosomal proteins, lipid-metabolic proteins, homeostatic molecules, integrins, pro-inflammatory cytokines, chemokines, MHC class I and MHC class II molecules for their important roles in the immune surveillance and maintaining homeostasis of CNS (Fig 2A). These included commonly reported homeostatic genes of microglia, including purinergic receptors (P2RY12 and P2RY13), fractalkine receptor (CX3CR1), selectin P (SELPLG), triggering receptor expressed on myeloid cells (TREM2), and tyrosine-protein kinase MER (MERTK). Micro-0 and Micro-1 had higher expression of these genes than the other four microglial clusters, indicating they were less likely to be associated with activation and inflammation. However, compared with Micro-0, Micro-1 had enhanced lysosomal activities and higher expression of some inflammatory molecules (APOE, CCL3, and CCL4), suggesting it might be slightly more activated or preactivated. Micro-3 highly expressed chemokines CCL3, CCL4, and MHC class I molecules, and Micro-5 had high expression of apolipoproteins (APOE, APOC1, and APOC4) and MHC class II molecules suggested that both Micro-3 and Micro-5 might be the activated, or response to infection, potentially disease-related microglia. Given that more inflammatory molecules and cytokines were highly expressed, and the homeostatic genes were rarely expressed in Micro-7 and Micro-11, those two microglial clusters might be in a higher activated state. However, Micro-11 had higher expression of MHC class II molecules with lower expression of homeostatic genes, suggesting it might be the most activated microglial population. Thus, in concert with our findings above, the proportion of homeostatic microglia decreases, whereas the proportion of activated microglia increases during acute SIV infection.

We then implemented GO analyses with the DEGs in each microglial cluster to assess biologic functionalities (Fig 2B and 2C). Micro-0 was identified with the pathways associated with transforming growth factor beta (TGFβ), which further confirmed the homeostatic status of this microglial cluster [33]. The other five microglial clusters were all active in the translation and biosynthesis, but Micro-3, Micro-5, Micro-7, and Micro-11 had additional pathways related to microglial activation, such as antigen-presenting ability in MHC class I or MHC class II manner and chemotactic ability. However, the activated microglia clusters appeared to specialize in different inflammatory or activation pathways, suggesting the heterogeneity of the activated or pathogenic microglial phenotypes in the brain. In summary, these results indicated that Micro-0 represented homeostatic microglia, Micro-1 might be a preactivated cluster with more activities in protein translation and lysosomal functions, and the other four clusters were the activated microglia with upregulation of different inflammatory molecules and pathways.

## Genes and pathways related to the MHC class I and type I interferon production were upregulated in all microglial clusters responding to acute SIV infection

In characterizing the microglial clusters above, we found one homeostatic cluster, one preactivated microglial cluster, and four activated clusters. Although the cells from uninfected

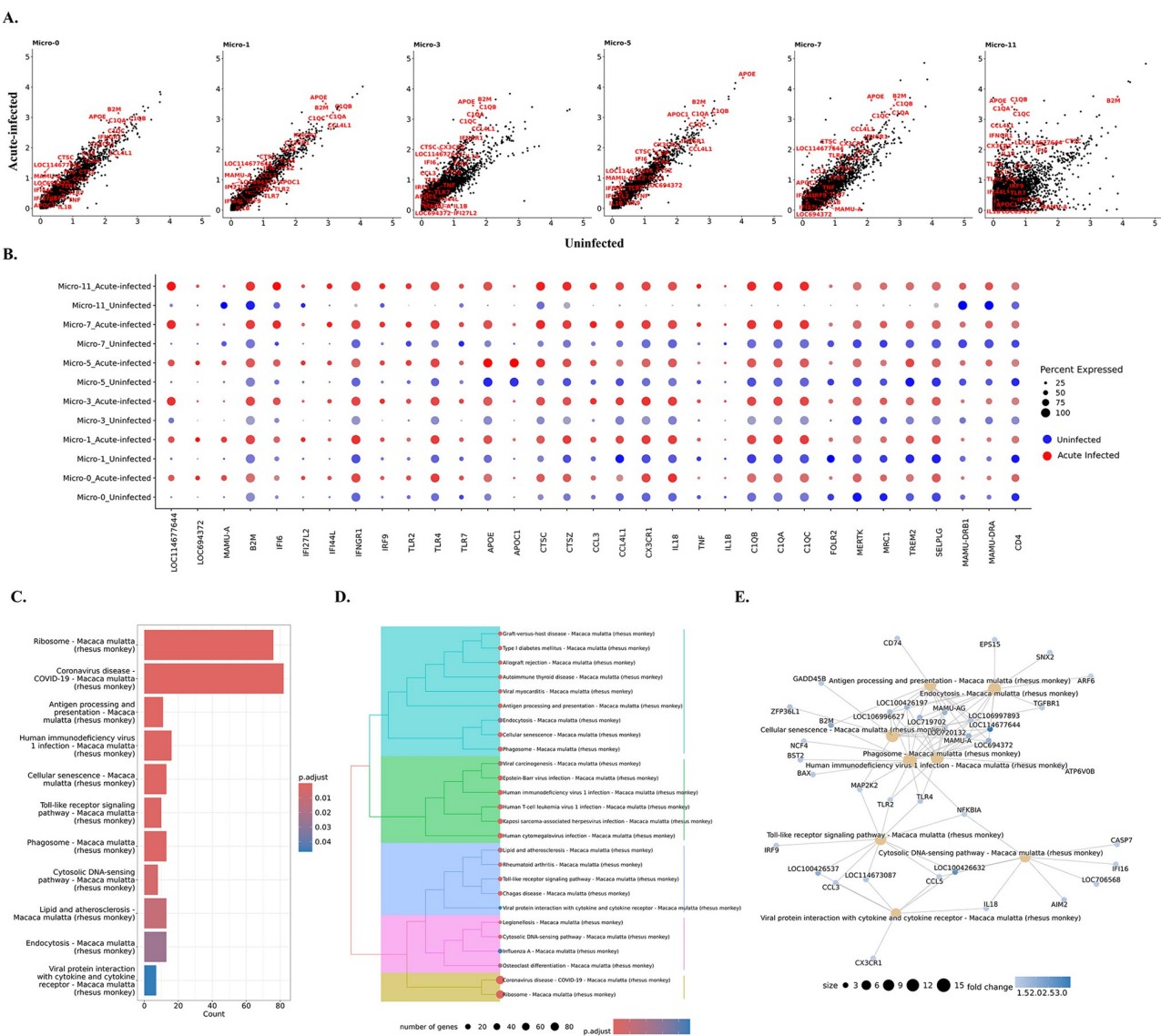

**Fig 3. Genes and pathways in microglia that were changed during acute SIV infection. (A)** The comparison of gene expression levels between the acute-infected animals and uninfected animals for each microglial cluster. Some of the upregulated genes were labeled in red. **(B)** Expression of selected genes that were differentially expressed in uninfected animals (blue dots) and animals with acute infection (red dots) for each microglial cluster. **(C)** Top upregulated pathways in microglia from acute infection animals. The upregulated genes of microglia during the acute SIV infection were used for the KEGG-ORA. **(D)** The hierarchical clustering of upregulated KEGG pathways for microglia in acute infection. **(E)** The gene-pathway networks between the specific pathways and human immunodeficiency virus 1. The complete pathways' analyses with counts of mapped genes and p values are shown in S5 Table. The MHC class I molecules are LOC114677644, LOC694372, LOC100426197, and LOC719702.

animals and infected animals were aggregated together for clustering, they still differ in the expression of some genes. Therefore, we identified the genes significantly upregulated in each microglial cluster due to acute SIV infection (S4 Table). More upregulated genes with high fold-change in acute SIV infection were found in activated clusters Micro-3 and Micro-11, where the average expression of the genes associated with activated microglia was low in uninfected animals (Fig 3A). Therefore, for those two clusters, only the microglial cells from infected animals have truly activated microglia. For the other activated or preactivated

microglial clusters, the cells from uninfected animals did have higher expression of the genes associated with activated microglia compared with the homeostatic cluster Micro-0, and those clusters further upregulated the expression of those activation genes in responses to acute SIV infection.

To better understand the changes in individual microglial clusters during acute SIV infection, we compared uninfected and acute infection samples for each microglial cluster (Fig 3B). The genes related to MHC class I and interferon (IFN) signaling pathways were universally upregulated in microglial clusters during acute infection. It is well-recognized that IFNs can block HIV/SIV replication [34–36]. This study found that the key transcription factor in the JAK-STAT pathway for IFN production, IRF9, was significantly upregulated in microglial cells. The increased fold-change of IRF9 ranged from 1.4 (Micro-11) to 8.3 (Micro-3), indicating this molecule was extensively upregulated in acute SIV infection. Furthermore, the genes encoding type I interferon-inducible proteins (IFI genes), especially IFI6, IFI27, and IFI44, were significantly upregulated in all microglial clusters. Those IFI genes are classified into interferon-stimulated genes (ISGs), which can be stimulated by the IFNs' signaling to augment the restriction of HIV/SIV replication and cell entry. In addition, we also found that the IFNγ receptor (IFNGR1) was slightly increased in most microglial clusters (~1.1-fold change) and highly increased in the Micro-11 cluster (4.5-fold change). Other upregulated genes included those encoding for apolipoproteins (e.g., APOE, APOC1), lysosomal proteins (e.g., CTSC, CTSZ), complement components (e.g., C1QA, C1QB, C1QC), chemokines (e.g., CCL3, CCL4, CX3CR1), and proinflammatory cytokines (e.g., IL18, TNF, IL1B). Although most of the aforementioned molecules were found to be upregulated in all microglial clusters, Micro-1, Micro-3, Micro-7, and Micro-11 were the clusters that had higher expression of those genes compared with Micro-0 and Micro-5 under acute SIV infection (S4A Fig). It should be noted that the Micro-5 cluster had the highest expression of APOE and APOC in the uninfected and acute infection conditions compared with other clusters (Fig 3B). APOE and APOC are well-described risk factors for numerous neurocognitive disorders (e.g., Alzheimer's disease) [37], and the APO ε4 variant was found to contribute to poorer cognition in PWH [38,39]. The significant upregulation of APOC and APOE in untreated HAND cases was also reported in another study using bulk-seq to find the risk molecules in HAND [40]. Therefore, the higher expression of apolipoproteins in the Micro-5 cluster for infected and uninfected conditions suggested its susceptibility to neurocognitive disorders regardless of SIV infection.

Surprisingly, unlike the upregulated MHC class I molecules, the MHC class II molecules remained unchanged or downregulated for most microglial clusters. The downregulation of MHC class II molecules was more obvious in Micro-7 and Micro-11, in which the MHC class II molecules had higher expression than other clusters (Fig 2A). The other downregulated genes were related to immunoregulation and microglial homeostasis (e.g., MERTK, TREM2, MRC1, FOLR2, SELPLG). This downregulation was significant in Micro-0, Micro-1, Micro-5, and Micro-7, but for Micro-3 and Micro-11 these genes' expression was upregulated. When we compared the expression of these immunoregulatory or homeostatic molecules between different microglial clusters in acute SIV infection, we found that their upregulation in Micro-3 leads this cluster to have higher gene expression than other clusters (S4A Fig). Intriguingly, CD4 gene expression was downregulated in all microglial clusters (fold-change ranging from 1.1–1.3) and had the highest expression in Micro-0 during the infection. CD4 serves as an important receptor for virus entry to the microglia [41] as well as the other targets of SIV/HIV infection, and its downregulation might indicate the defensive strategy of activated microglia in acute SIV infection. While CD4 downregulation by HIV and SIV infection of cells is known to occur through viral accessory proteins targeting the CD4 protein for degradation, [42] the

downregulation of its transcript in all of the microglial clusters in acute infection likely results from cellular activation.

The genes significantly upregulated in microglial cells were then further associated with various pathways using the Kyoto Encyclopedia of Genes and Genomes (KEGG) databases (Fig 3C–3E). All six clusters upregulated pathways related to antigen presentation, processing, and cellular senescence. Micro-0 was found to upregulate the least number of pathological pathways compared with other microglial clusters (S5 Table), which suggests that the homeostatic status in Micro-0 was least altered by acute SIV infection. The other five microglial clusters were associated with more disease-related pathways under acute infection (S3 Fig). In general, these pathways could be categorized into five classes (Fig 3D), including pathways related to antigen processing and presentation, viral infections, inflammation, cytosolic DNA sensing, and ribosomal activities. In the category of viral infections, we found the HIV-1 infection pathway with sixteen enriched genes. Most genes upregulated in the HIV-1 infection pathway encoded MHC class I molecules, which connected the endocytosis, phagocytosis, and antigen presentation pathways to HIV-1 infection (Fig 3E).

The MEK2 protein kinase (MAP2K2) molecule, involved in the mitogen-activated protein kinase (MAPK) pathway, also serves as the key molecule connecting cellular senescence and toll-like receptor (TLR) signaling with HIV-1 infection. MAP2K2 can trigger inflammation by phosphorylating the downstream kinases ERK1/2 (Extracellular Signal-Regulated Kinases 1 and 2) to translocate the transcription factor NF-κB and AP-1 to the nucleus for the expression of the genes encoding cytokines. The TLRs utilize those pathways to induce the production of proinflammatory cytokines. By comparing different TLRs, we found that TLR4, which can induce the expression of type I IFNs, was upregulated in all microglial clusters (Fig 3C) during acute SIV infection. TLR2 signaling was upregulated in most microglial clusters, which acted through NF-κB and AP-1 to produce proinflammatory molecules (such as TNF-α, IL-1β, and IL-6). TLR7 which, as opposed to the cell-surface location of TLR4 and TLR2, is located on the endosomal compartment of the cells and can specifically recognize single strand RNA (ssRNA) of HIV/SIV for the production of type I IFNs [43–45], was found to be upregulated particularly in Micro-3 (2.8-fold change). While the TLRs were extensively upregulated in microglial cells, Micro-7 and Micro-11 had the highest expression of TLR2, Micro-1 had the highest expression of TLR4, and Micro-3 and Micro-7 had the highest expression of TLR7 during acute SIV infection (S4A Fig), indicating different microglial clusters might favor different TLR signaling pathways to produce proinflammatory cytokines. In summary, all of the upregulated pathways in acute SIV infection pointed to the pathways related to interferons and TLR-induced inflammatory cytokine production, highlighting their critical roles in microglia defense against acute SIV infection.

## The predominant CAM cluster phenotype in acute SIV infection was CD14$^{hi}$CD16$^{low}$

In characterizing the predicted phenotypes for those two macrophage clusters, we found that Macro-6 had lower expression of the inflammatory molecules highly expressed in Macro-8 (e.g., APOE, CST3, MSR1). Instead, it had higher expression of the cell adhesion molecules (e.g., PECAM1 and integrins) (Fig 4A and S2 Table). In addition, Macro-6 was found to have a high expression of CD16 (FCGR3) but a low expression of CD14, and Macro-8 had a high expression of CD14 but a low expression of CD16 (Fig 4B). In human blood, CD14$^{hi}$CD16$^{low}$ cells are described as inflammatory classical monocytes/ macrophages that are trafficked to sites of inflammation and/or infection, and CD14$^{low}$CD16$^{hi}$ cells are the patrolling non-classical monocytes/macrophages, which adhere and crawl along the luminal surface of endothelial

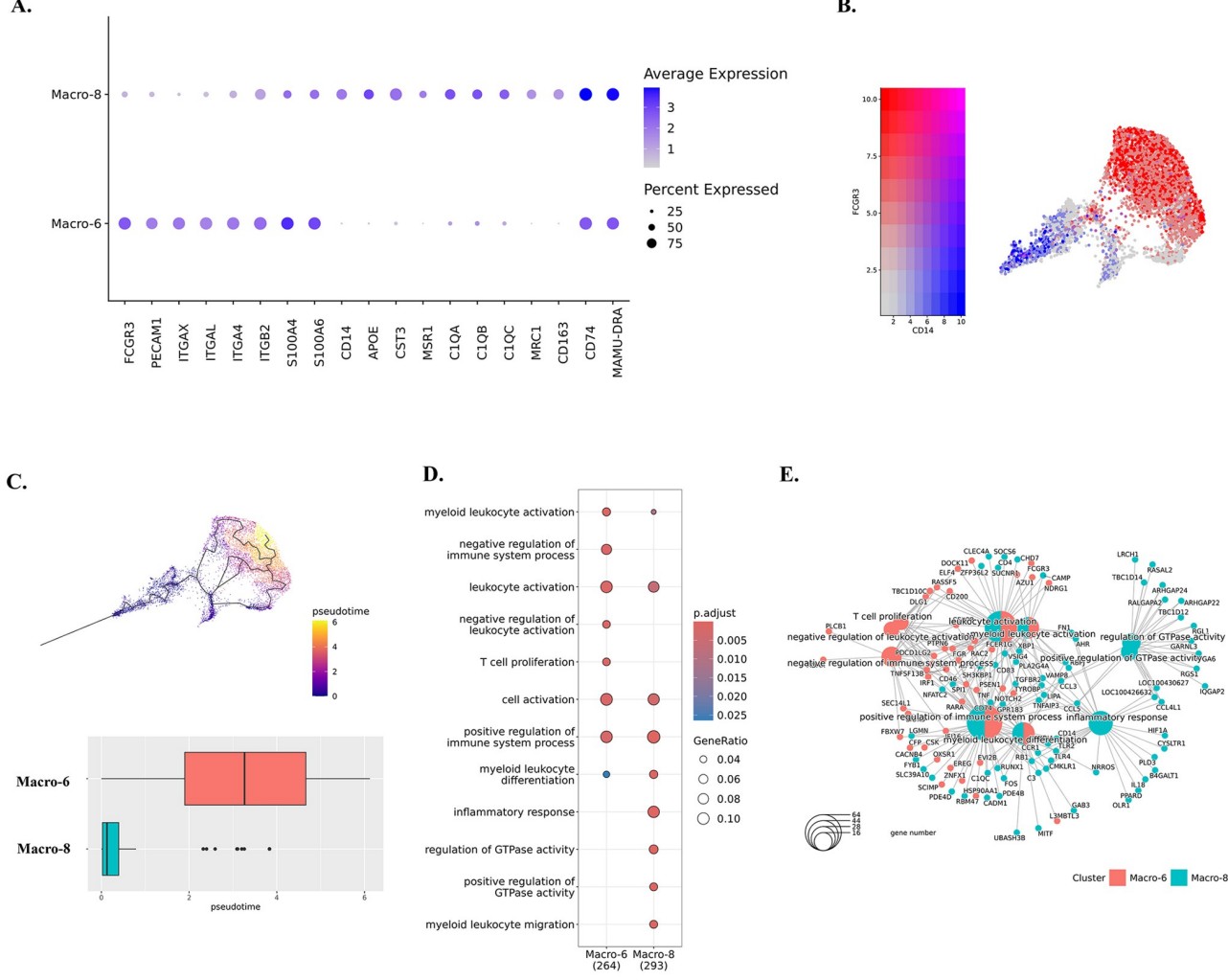

**Fig 4. Characterizations of identified macrophage clusters. (A)** Expression of DEGs in Macro-6 and Macro-8. The DEGs of each macrophage cluster were found by using Wilcoxon rank sum test for comparison. **(B)** The expression of CD14 and CD16 (FCGR3) in macrophage clusters. **(C)** The trajectory analyses for Macro-6 and Macro-8. The cell map indicated the trajectory paths, and the comparison of pseudotime between Macro-6 and Macro-8 was shown in the boxplot. **(D)** The characteristic biological processes in each macrophage cluster by Gene Ontology (GO) pathway analyses. **(E)** The networks between genes and biological pathways for macrophage clusters.

cells [46]. In this study, we found that acute SIV infection shifted the dominant CAM phenotype from CD14$^{low}$CD16$^{hi}$ (Macro-6) to CD14$^{hi}$CD16$^{low}$ (Macro-8). Although Macro-8 appears to represent the classical inflammatory phenotype, it also had higher expression of the markers for anti-inflammatory M2-like macrophages (e.g., MRC1/CD206, CD163). Regarding differentiation, it was reported that the CD14$^{hi}$CD16$^{low}$ phenotype can be directly differentiated from precursor cells in bone marrow, and it is the obligatory precursor intermediate for the CD14$^{low}$CD16$^{hi}$ phenotype in the blood [47]. Intriguingly, through trajectory analyses, we also found that the CD14$^{low}$CD16$^{hi}$ CAMs (Macro-6) had a larger value of pseudotime compared to the CD14$^{hi}$CD16$^{low}$ CAMs (Macro-8) (Fig 4C), suggesting CD14$^{hi}$CD16$^{low}$ cells might also be the precursor intermediate for CD14$^{low}$CD16$^{hi}$ cells in CNS.

The GO results indicated that cells in the Macro-6 and Macro-8 clusters participated in myeloid leukocyte activation and differentiation, suggesting these two phenotypes might

originate or differentiate from infiltrating monocytes. According to the opposite biological functions of CD14$^{hi}$CD16$^{low}$ and CD14$^{low}$CD16$^{hi}$ monocytes in the periphery, the Macro-6 cluster had positive and negative immune process regulations. However, the Macro-8 cluster lacked the negative regulation of the immune responses and thus had the potential to trigger inflammation (Fig 4D). Furthermore, those two CAM clusters might also have the potential for interactions with infiltrating lymphocytes and adaptive immune responses, functions that were not found in the microglia clusters. For example, Macro-6 was predicted to regulate lymphocyte proliferation by secreting IL-15, an important stimulator for T/NK cells' proliferation and activation. Macro-8 highly expressed MHC class II and other molecules can present the antigens for triggering adaptive immune responses (Fig 4E). In summary, we identified both CD14$^{hi}$CD16$^{low}$ and CD14$^{low}$CD16$^{hi}$ macrophage phenotypes in the brains of the rhesus macaque, and the CD14$^{hi}$CD16$^{low}$ cells (Macro-8 cluster) became the dominant phenotype in acute SIV infection, whereas CD14$^{low}$CD16$^{hi}$ cells (Macro-6 cluster) predominated in the uninfected condition.

## CAMs might be more activated in triggering immune responses under acute SIV infection than microglia

Consistent with the genes upregulated in all microglial clusters, most molecules related to MHC class I and IFN production were also significantly upregulated in both macrophage clusters (Fig 5A and 5B). However, unlike microglial clusters, the IFI44L gene had extremely low expression in macrophage clusters. The MHC class II molecule changes in macrophage clusters differed between Macro-6 and Macro-8. The expression of MHC class II molecules in Macro-6 remained unchanged except for the downregulation of MAMU-DPA. However, the expression of those genes in Macro-8 was extensively downregulated in acute SIV infection, which is consistent with the changes in activated microglial clusters. Intriguingly, we also found that the CAMs significantly upregulated the core genes for homeostatic microglia during acute SIV infection (Fig 5B). Given the myeloid lineage of microglia and CAMs, similar behaviors in response to acute SIV infection might be expected (e.g., APOE and SPP1), but it was surprising that the homeostatic genes for microglia (e.g., P2RY12, GPR34) and other genes characterizing microglia (MERTK, SELPLG) that were downregulated or remained unchanged in microglial clusters during acute SIV infection were increased in CAMs, especially Macro-6. We found that P2RY12 expression was upregulated 4-fold, GPR34 4.5-fold, SELPLG 2.4-fold, and MERTK 3.7-fold in the Macro-6 cluster. Although Macro-6 highly upregulated those molecules, the expressions of microglial homeostatic core genes were still higher in Macro-8 during acute infection (S4B Fig). Furthermore, while Macro-6 was characterized as CD14$^{low}$CD16$^{hi}$ macrophages, and Macro-8 was characterized as classic CD14$^{hi}$CD16$^{low}$ macrophages, once they were in acute infection condition, the CD14$^{hi}$CD16$^{low}$ cells upregulated CD16 (1.5-fold), whereas the CD14$^{low}$CD16$^{hi}$ cells upregulated CD14 (3.3-fold). This again highlights the plasticity of myeloid cells.

The elevation of CD14 in Macro-6 was also accompanied by the increased expression of TLR4, which is the coreceptor for CD14 in inducing pro-inflammatory signaling. In addition, more genes related to the inflammatory pathways (e.g., TLR2, TLR9, IRF8, IRF9, C1QA, C1QB, C1QC) were found to be increased in both macrophage clusters (Fig 5B). The fold-change of most inflammatory molecules was higher in Macro-6 (e.g., C1QA, C1QB, and C1QC had a 2-fold change in Macro-6 but only ~1.2-fold change in Macro-8) but the overall expression level was still higher in Macro-8 compared to Macro-6 (S4B Fig). Macro-8 also significantly upregulated more molecules related to innate defense, such as S100A8 (fold-change: 2.3) and S100A9 (fold-change: 2.9), which was not observed in Macro-6. The

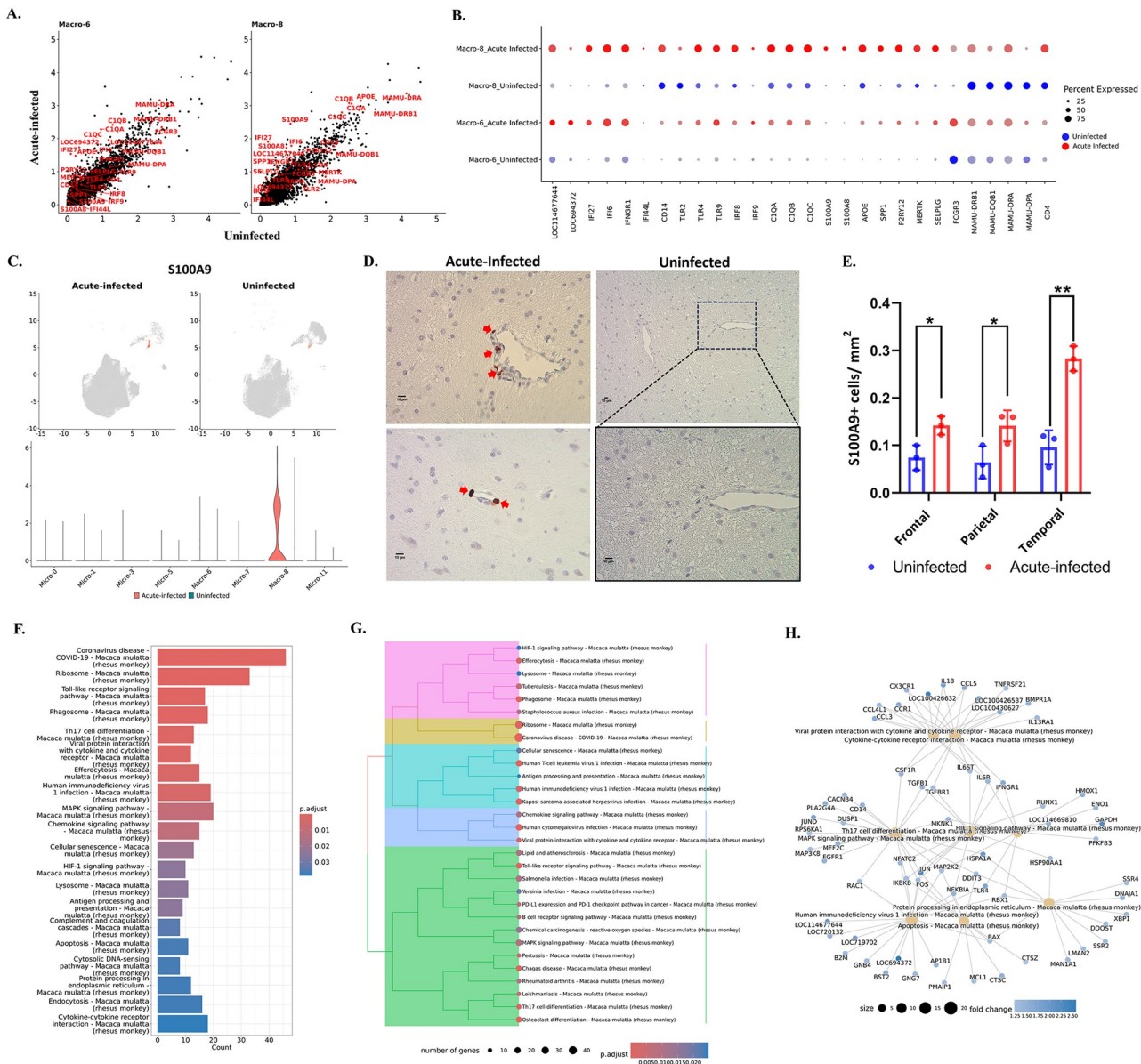

**Fig 5. Altered gene expressions and pathways in macrophage during acute SIV infection. (A)** The comparison of the gene expression levels between the acute-infected animals and uninfected animals for macrophage clusters. Some of the significantly changed genes were labeled in red. **(B)** Selected DEGs in uninfected animals (blue dots) and animals with acute infection (red dots) for each macrophage cluster. **(C)** The expression of S100A9 in brain myeloid cell clusters. The acute-infected and uninfected conditions were plotted separately. **(D)** Representative images (scale bars, 10 μm) of S100A9 staining in the parietal (upper) or temporal (lower) lobes from frontal of acute-infected animals (left) and temporal lobe of an uninfected animal (right). The red arrow indicates the S100A9+ cells (cells in the Macro-8 cluster) found in the perivascular space of the acute-infected animals. **(E)** Quantification of S100A9+ cells per mm² in different brain regions. Adjusted p value: * < 0.05, ** < 0.01. **(F)** Top 20 upregulated pathways in macrophages from acute infection animals. Overall upregulated genes with adjusted p values of less than 0.05 in macrophages were used for the Kyoto Encyclopedia of Genes and Genomes (KEGG) pathways analyses. **(G)** The hierarchical clustering of upregulated KEGG pathways of macrophage in acute infection. **(H)** The gene-pathway networks between the specific pathways and human immunodeficiency virus 1 pathway. LOC114677644 and LOC694372 are the MHC class I molecules.

immunohistochemistry staining of the Macro-8 marker, S100A9 (Fig 5C), further confirmed that there was a significant increase S100A9+ myeloid cells (Macro-8), notably in the perivascular space (Fig 5D) of several brain regions during acute SIV infection (Fig 5E). In summary, these results suggest that the immunosuppressed phenotype of Macro-6 might differentiate toward an inflammatory phenotype during the acute infection.

The KEGG gene enrichment analyses for macrophages showed more upregulated inflammatory pathways compared to microglia (S3 Fig and S5 Table). Overall, the macrophages not only enhanced the pathways that were found to be upregulated in microglia, but also augmented other inflammation-related pathways, for example, the interaction between viral proteins and cytokine/chemokine receptors and Th17 cell differentiation (Fig 5F and 5G). In the characterization of the CAMs by their featured pathways, we found that these macrophages might have more interactions with the T cells than microglial cells. When we further examined the genes that were upregulated during acute SIV infection in the KEGG pathways, we found that they were enriched in the ability to induce Th17 differentiation during acute SIV infection. The genes that are mapped to this pathway included IL-6 and TGFB (Fig 5H), which were reported as key cytokines secreted by macrophages to trigger differentiation of Th17 cells [48,49]. Given the pleiotropic functions of those two cytokines, their increase might not be necessarily correlated with Th17 differentiation, but their increased expression in perivascular macrophages responding to acute SIV infection has been confirmed in rhesus macaque [50]. Even though more inflammatory or immunological pathways were upregulated in macrophages compared to microglia, the key pathway that connects them with HIV-1 infection might also involve the NF-κB signaling. In the gene-pathway connections (Fig 5H), multiple molecules and kinases for NF-κB signaling (e.g., IKBKB, JUN, FOS, MAP2K2, NFKBIA, TLR4) connected the HIV-1 infection and other inflammatory or anti-viral pathways.

## The genes upregulated in Micro-3 and Macro-8 clusters in response to acute SIV infection are associated with multiple neurological diseases

The microglial and macrophage clusters that were widely activated, with increased expression of proinflammatory cytokines and chemokines in acute SIV infection, might induce toxicities not only for the virus but also for neurons and other cells in the CNS. Here, we found that one microglial and one macrophage cluster may be strongly related to neurological disorders such as HAND. The upregulated genes analyzed with KEGG gene enrichment analyses for each cluster revealed pathways leading to multiple neurocognitive disorders in both the Micro-3 and Macro-8 clusters (Fig 6A). Prion disease, Parkinson's disease, Alzheimer's disease, Huntington's disease, and amyotrophic lateral sclerosis were in the list of this category. In Micro-3, all of those five neurodegeneration-associated pathways have extremely low adjusted p-values (<0.002); in Macro-8, Huntington's disease and amyotrophic lateral sclerosis had slightly higher adjusted p-values (~0.006). The mapped genes in those pathways were related to mitochondrial functions, and we found that the fold-change of the associated genes in Macro-8 was higher than in Micro-3 (Fig 6B).

To further confirm the specificity of neurodegenerative molecules in Micro-3 and Macro-8 during the acute SIV infection, we then compared the expression changes for a number of these genes in all microglia and macrophages under uninfected conditions and infected conditions. Consistent with the above, we found that those genes were significantly upregulated in Micro-3 and Macro-8 in acute SIV infection (Fig 7A and 7C). However, the expression of these genes barely changed when we compared them in all microglial cells (Fig 7B). Although genes that are related to the mitochondria functions seemed to be upregulated when all

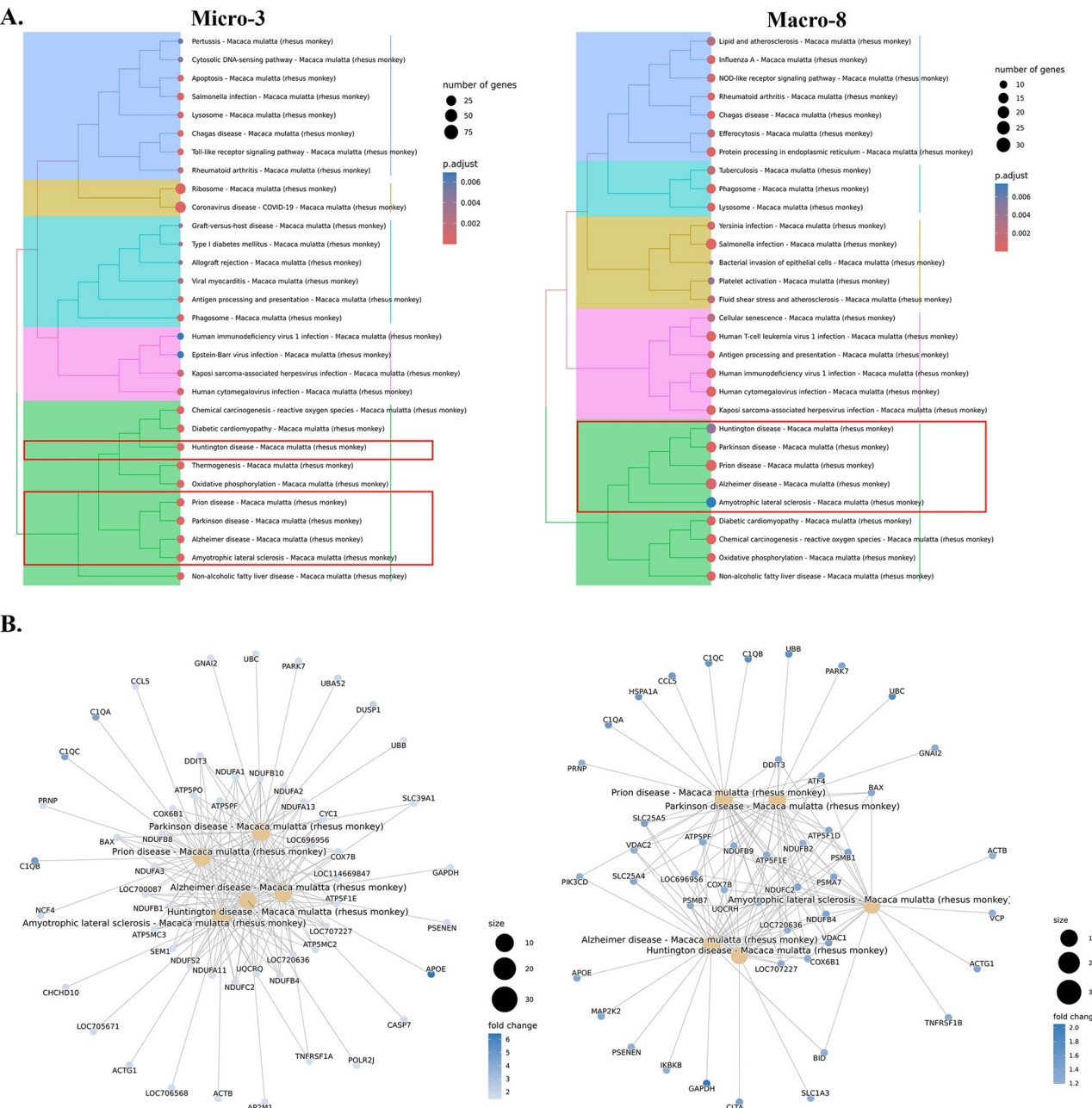

**Fig 6. Potential neurodegenerative pathways associated with Micro-3 and Macro-8. (A)** In the hierarchical clustering of the upregulated KEGG pathways for each cluster during the acute SIV infection, there was a category for Micro-3 (left) and Macro-8 (right) enriched with multiple neurodegenerative diseases. There were five neurological diseases identified in both clusters, including Prion disease, Parkinson's disease, Alzheimer's disease, Huntington's disease, and Amyotrophic lateral sclerosis. **(B)** The genes associated with each mapped the neurodegenerative pathway for Micro-3 (left) and Macro-8 (right).

macrophages in infected and uninfected conditions were compared (Fig 7D), the overall changing fold was not as prominent as in the Macro-8 cluster. All of these results suggested that the Micro-3 and Macro-8 clusters might be strongly associated with the induction of neurological disturbances when activated in acute SIV infection.

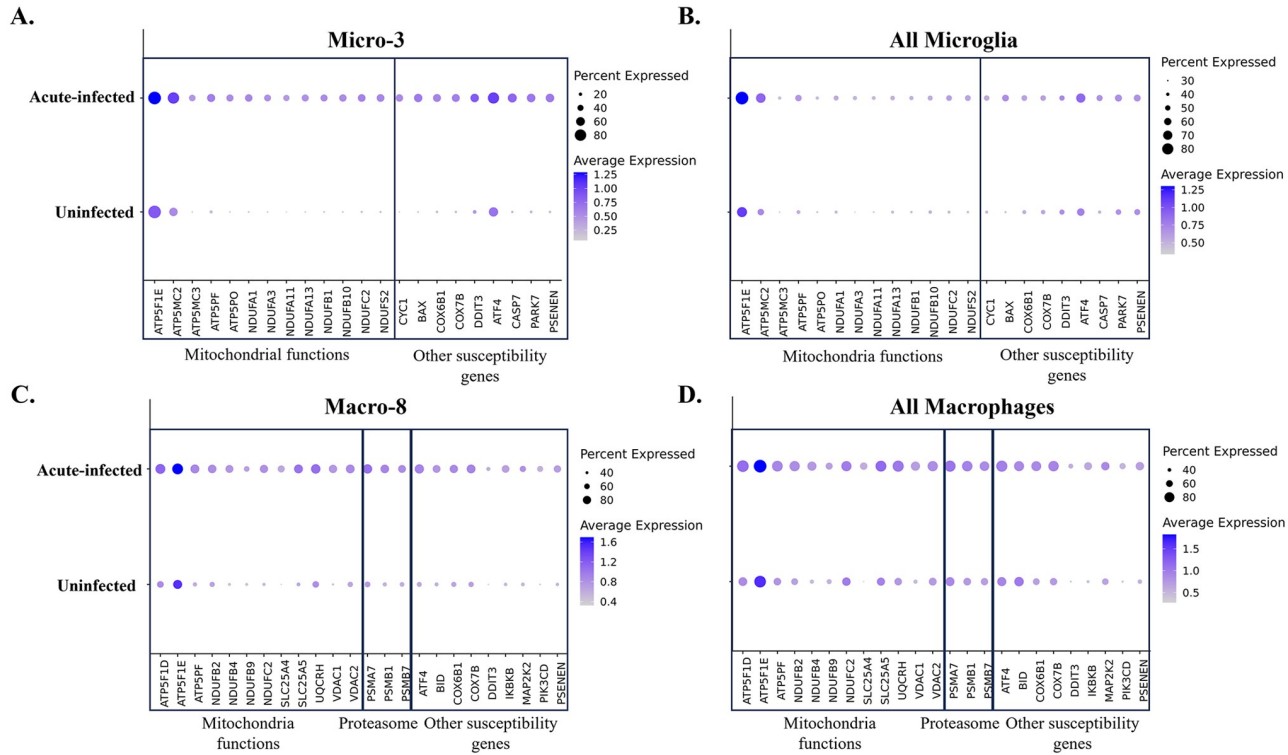

**Fig 7. Comparison of the susceptibility genes for neurodegenerative disorders in uninfected animals and acute-infected animals.** The comparison was conducted for the cells in **(A)** Micro-3 cluster, **(B)** all microglial clusters, **(C)** Macro-8 cluster, **(D)** and all macrophage clusters separately. The susceptibility genes mapped to the neurodegenerative pathways were compared, which includes the genes related to the mitochondria and proteasome functions.

## Apoptotic resistance was not initiated during acute SIV infection in myeloid cells in the brain

HIV was reported to upregulate the expression of anti-apoptotic molecules and downregulate pro-apoptotic molecules to enable the survival of infected host cells [28,29,51,52]. Those anti-apoptotic molecules (e.g., BCL2, BFL1, BCL-XL, MCL-1) and pro-apoptotic molecules (e.g., BAX, BIM, BAK1, BAD) are typically found in the BCL-2 family. While microglia and macrophages are typically thought to be long-lived, the mechanism of survival following HIV infection has not been extensively studied. To examine if HIV infection of myeloid cells in the brain could lead to the establishment of the virus reservoir by blocking apoptosis, we first compared the expression of anti-apoptotic and pro-apoptotic molecules between macrophages and microglia in acute-infected animals and uninfected animals (Fig 8A). However, we did not observe remarkable changes in the RNAs encoded by these molecules in macrophages and microglia during acute infection. We did observe that the anti-apoptotic molecule BCL2 was downregulated, and the pro-apoptotic molecule BAX was upregulated, suggesting that the macrophages and microglia are prone to undergo apoptosis *in vivo* upon SIV infection. Then we further investigated the change of these molecules in individual myeloid cell cluster (Fig 8B). The BCL2 gene was downregulated in most clusters, and the cluster which did not downregulate BCL2 (Macro-6) was found to maintain its expression without change. On the other hand, the pro-apoptotic gene, BAX was widely upregulated in most cell clusters. For other anti-apoptotic gene expressions, we found that MCL1 was upregulated, except in homeostatic

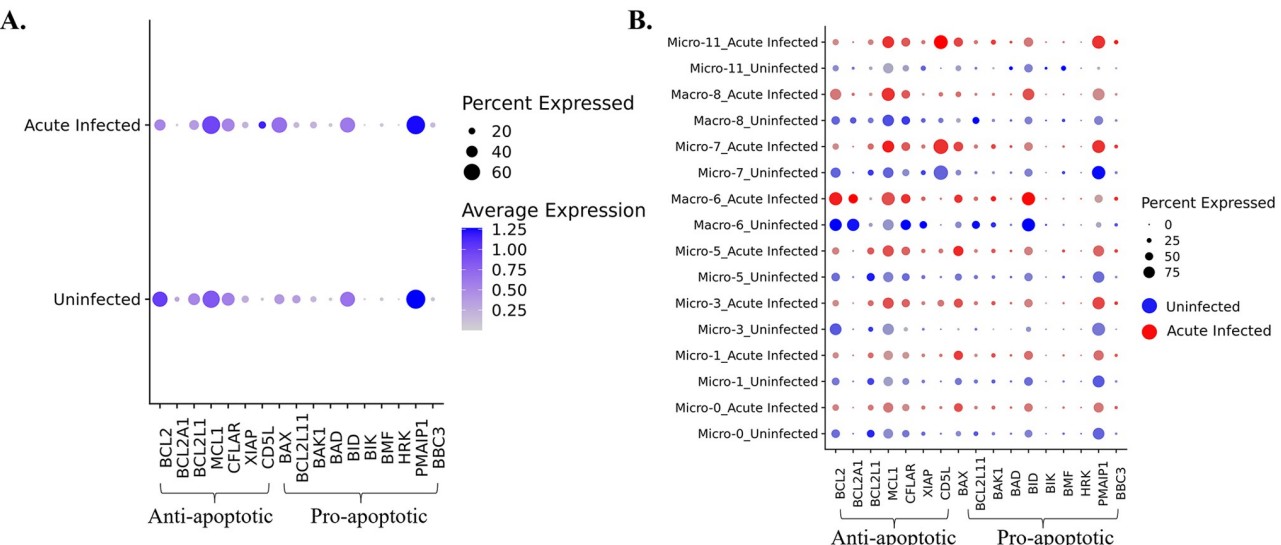

**Fig 8. Changes of anti-apoptotic and pro-apoptotic genes in brain myeloid cells due to acute SIV infection.** **(A)** The average expression of anti-apoptotic and pro-apoptotic genes in acute infection or uninfected animals was compared in all macrophages and microglia combined. **(B)** The average expression of anti-apoptotic and pro-apoptotic genes in acute infection (red dots) or uninfected (blue dots) animals was compared in each macrophage and microglial cluster.

Micro-0 and Micro-1, CFLIP (CFLAR) was upregulated in Micro-3 and Micro-11, and others remained unchanged or slightly downregulated in infection. The pro-apoptotic molecules were generally upregulated, for example, BAK1 was upregulated in Micro-3, Micro-7, and Micro-11, BID was upregulated in Micro-3 and Macro-8, and PMAIP1 was upregulated in Micro-11. However, a known pro-apoptotic molecule for microglia and macrophage, BIM (BCL2L11), was found to be downregulated in Macro-6 and Macro-8. Interestingly, we found another anti-apoptotic molecule, CD5L, which is not included in the BCL-2 family, was highly upregulated in Micro-11.

## Discussion

Microglia are the major innate immune cells in the brain and serve multiple functions. They protect the brain from invading pathogens, clear damaged synapses and dying cells, and promote neuron development. While not as extensively characterized as neurons, they are indeed heterogeneous, and with the advent of scRNA-seq, our studies and those of others have identified several classes of microglia [53–56]. Microglia express several homeostatic genes to maintain a steady state while gaining some immunogenic properties. These homeostatic genes' downregulation and/or mutations can lead to uncontrolled neuroinflammation, neurotoxicity, and multiple neurocognitive diseases [57–61]. In this study, we found multiple populations of microglia that changed their proportions during acute SIV infection. During acute SIV infection, overall activation was observed, with a lower expression of homeostatic genes and a decreased proportion of cells with a homeostatic phenotype. In addition, we found that the major homeostatic microglial cluster, Micro-0, was enriched in the TGFβ signaling pathway, which was not observed for the other microglial clusters. Based on others' findings, the silencing of TGFβ signaling in microglia resulted in the loss of microglial ramification and the upregulation of inflammatory markers without any external stimulus relative to wild-type microglia [33,62]. Considering the pleiotropic functions of TGFβ signaling in microglial cells, [63–65] the more active TGFβ pathway in Micro-0 suggested that this homeostatic microglial

cluster might be more active in stimulating microglial differentiation and regulating the activation of other microglial clusters.

Although the proportion of cells in Micro-0 remarkably decreased during acute SIV infection, this was still the dominant microglial population (consisting of 44.1% of microglial cells in acute infection, compared to 64.1% in uninfected animals) in the brain, suggesting that microglial activation was still under some control during the acute infection stage. In addition, Micro-0 did not show prominent signs of activation during this stage, which could be attributed to their high activity in TGFβ signaling. The Micro-1 cluster, with high microglial core gene expression but lacking TGFβ signaling pathway enrichment, was much more inducible regarding gene expression during acute SIV infection than the homeostatic Micro-0 cluster. More pathogenic and inflammatory pathways were upregulated for Micro-1 during acute SIV infection (S3 Fig), highlighting the importance of the TGFβ signaling pathway in regulating microglial activation.

Micro-1, similar to Micro-0, also decreased its proportion in acute infection, comprising 27.1% of the myeloid cells versus 32% in the uninfected animals. These decreases were accompanied by prominent changes in the proportions of other microglial populations in acute infection, as Micro-3, -5, -7, and -11 all increased their representation. For CAMs, the dominant cluster changed from Macro-6 (65% in uninfected animals) to Macro-8 (72.9% in acute SIV infection). All these were characterized by increased signs of cellular activation. By assessing S100A9+ cells in brain sections we found that S100A9+/Macro-8 cells increased in acute SIV infection in multiple regions. While we only examined one cell cluster related to acute infection, we hypothesize that changes of the other myeloid cell phenotypes can also occur throughout the brain. Furthermore, given that microglia in different brain regions may respond to SIV/HIV protein and likely infection differently, [66] future studies can address whether regional differences exist.

In uninfected brains, the activated microglial clusters constituted a low but noticeable presence (<5% proportion). Those activated microglia cells have been reported to exist in the healthy brain at different anatomic locations and are involved in diverse neurological events compared with homeostatic microglia [67–69]. From scRNA-seq experiments, it is difficult to determine the biological meaning of such low-proportion cell clusters. Therefore, to better interpret the changes in gene expression during the acute SIV infection, we made the comparisons in multiple ways.

We found that MHC class I genes and genes related to interferon production were significantly upregulated in both microglial and macrophage clusters during acute SIV infection. However, the expression levels differed between individual clusters, suggesting discrepancies in responses for each cluster. For virus infection, MHC class I molecules are key to the host defense for their ability to present virus proteins in the infected cells to the cytotoxic T/NK cells in MHC class I manner [70]. The significant upregulation of MHC class I molecules in all myeloid cell clusters also indicated that the reaction to initial SIV infection might not be limited to the low proportion of microglia and macrophages infected with SIV and that the bystander cells also showed reactivity to the infection. This was further supported by the transcriptional changes in these bystander cells (S3 Table).

The expression of type I interferons, especially IFNβ [71,72] is widely upregulated in acute SIV infection. In response to the production of IFNβ, the downstream ISGs that are induced by type I IFNs to amplify the antiviral effects have been found to be upregulated in blood and lymph nodes during acute SIV/HIV infection in rhesus macaque and humans [73,74]. Our recent and prior studies of microglial responses to acute and chronic SIV infection in the brain [56,75] also highlighted that numerous ISGs were significantly upregulated in the myeloid cells of the brain and might contribute to HIV/SIV-associated brain damage. Therefore, the

high expression of ISGs starting at an early stage of SIV infection can extend to chronic infection in the brain's myeloid cell populations.

We note that one of the important pathways for type I IFN production is TLR signaling, and we found that TLR2 and TLR4 were widely upregulated in microglial and CAM clusters. TLR7, which can specifically bind with ssRNA from SIV, was upregulated in Micro-3, Micro-5, and Micro-11, and TLR9, which can bind with the CpG motif in DNA, was widely upregulated in CAM clusters. All of these TLRs were reported to be able to recognize and bind with HIV [72,74], although TLR7 might be the primary target [43]. The presence of TLR4, TLR7, and TLR9 in endosomal compartments endows them to recognize viral nucleic acids in the cells, and indeed their signaling pathways are related to the production of IFNs [76–78]. TLR signaling can trigger the transcription of numerous inflammatory cytokines (e.g., TNF-α, IL-6, IL-1β, etc), which were also upregulated in microglial or macrophage clusters under acute SIV infection. All of those cytokines can serve as initiator or the products for the pathways associated with NF-κB and AP-1 signaling, which is also modulated by HIV/SIV for their establishment and reactivation from latency [79–83].

CNS-associated macrophages in the interface between the parenchyma and the circulation represent an important myeloid cell population in the brain distinct from microglia. Previously, all CAMs were thought to be derived from the monocytes in the circulation, but recent findings showed that CAMs are highly heterogeneous. Some phenotypes might originate from the yolk sac and can be self-replenishable, like microglia, which differ from the CAMs, differentiating from the circulating monocytes. From the transcriptional perspective, the yolk sac-derived CAMs also share more similarities with microglia, and they are hard to separate based on their transcriptomic profiles [84]. Given the distance between CAM and microglial clusters on the UMAP and the low expression of microglial core genes in CAM clusters, Macro-6 and Macro-8 were more likely to be derived from the circulating monocytes. Like the two main phenotypes in the periphery, Macro-6 had a $CD14^{low}CD16^{hi}$ phenotype with immunosuppressive properties and Macro-8 had a $CD14^{hi}CD16^{low}$ phenotype with proinflammatory properties. During acute SIV infection, Macro-6 upregulated more inflammatory molecules already highly expressed in Macro-8, indicating the Macro-6 could be activated and polarized toward the inflammatory $CD14^{hi}CD16^{low}$ phenotype. This observation is consistent with a report about the transcriptomic convergency of the peripheral $CD14^{++}CD16^{+}$ and $CD14^{+}CD16^{++}$ population under the SIV infection [85].

Intriguingly, macrophages also increased the expression of microglial core genes during the acute SIV infection. In a different system, it was found that the monocyte-derived macrophages in the retina adopted microglia-like gene expression during the degeneration processes, [86] which was similar to what we found for the CAMs in acute SIV infection. However, the reasons CAMs share more similarities with microglia in response to stress or neurodegenerative diseases are still unknown. Although the CAM clusters synchronized the changes for most immunoreactive molecules, other immunoreactive molecules might change differently. For example, most MHC class II molecules that were not changed or slightly upregulated in Macro-6 were downregulated in Macro-8. The repression of MHC class II molecules in professional antigen-presenting cells (APCs) was also observed during HIV infection in humans and serves as one of the immunodeficiency mechanisms of CD4+ T cells in AIDS [87–90]. Therefore, the unchanged MHC class II in Macro-6 suggested that this cluster might be less likely to be affected by SIV-induced immunodeficiency, at least in acute infection. They may still maintain the ability to trigger the activation and differentiation of CD4+ T cells for host defense against the SIV virus.

Activating microglia and CAMs in the brain is essential for protecting against viral infection. However, the pro-inflammatory cytokines and cytotoxic molecules they secreted might

also damage the CNS's neurons and other supportive apparatus. Therefore, overreactive microglia, especially monocyte-derived CAMs, were thought to play a central role in HAND before the era of efficacious antiretroviral therapy [27,91]. By enriching the upregulated genes in each microglial or macrophage cluster into the various pathways, we found that Micro-3 and Macro-8 clusters were linked with numerous neurological diseases, including Alzheimer's, Huntington's, and Parkinson's diseases during acute SIV infection. These changes in CAMs and microglia also have potential in the pathogenesis of HAND by various disease mechanisms associated with neurodegeneration [27]. When we further identified the upregulated genes linked with those neurological disorders, we found that the genes related to mitochondrial respiration were significantly upregulated in both Micro-3 and Macro-8. *In vitro*, HIV infection of macrophages has been shown to alter their mitochondrial energetic profiles, with the specific changes dependent on the stage of infection [92].

Given the unmet need for the validated biomarkers of HAND in clinical settings, [93] some studies have sought to identify various proinflammatory cytokines and molecules associated with protein misfolding as biomarkers for HAND, [94,95] and other studies explored the specific myeloid cell phenotypes for serving as biomarkers [91,96]. Both HIV itself and the antiretroviral drugs used to treat HIV can affect mitochondria, and mitochondrial pathways are key suspects in the resulting neuropathogenesis of HIV and HAND [97].

Although only Micro-3 and Macro-8 showed enrichment in multiple neurological disorder pathways, all the myeloid cell clusters were associated with the cellular senescence pathway, which might be another potential factor triggering or dampening the neurological disturbances [98,99]. The potential effect of cell aging caused by acute SIV infection might happen in microglia or macrophages and extend to neurons and other brain cells. Indeed, while very few myeloid cells were directly infected, the bystander effects of infection were widely manifested. Furthermore, the inflammatory microglia and CAMs resulting from virus infection will likely trigger or speed up cerebral aging [100]. Indeed, epigenetic studies have found that HIV infection leads to advancing predicted biological age in blood cells and the brain and is associated with reduced brain gray matter and cortical thickness [101–104].

Both HAND and the establishment of HIV/SIV reservoir in CNS can occur irrespective of antiretroviral therapy [105,106]. Macrophages and microglia cells are believed to serve as major HIV reservoirs in the CNS, given that they are prime targets for the virus and their known longevity [17]. Many pathways could be altered by HIV infection, allowing for the long-term survival of the infected macrophages and microglia, and the anti-apoptotic or pro-apoptotic molecules in BCL-2 families were found to regulate the survival of macrophages and/or microglia in HIV infection [28,29,107,108]. However, we did not observe a wide and significant upregulation of anti-apoptotic molecules or downregulation of pro-apoptotic molecules in myeloid cell populations identified in this study. The cluster that upregulated anti-apoptotic genes also upregulated pro-apoptotic genes, so it is difficult to identify a specific cluster that might facilitate the establishment of an SIV and, by analogy, an HIV reservoir. From the average expression of the anti-apoptotic and pro-apoptotic molecules in BCL-2 families, the overall trend for the myeloid cells of the brain under acute SIV appears to be the enhancement of pro-apoptosis and suppression of anti-apoptosis and how reservoir cells survive long-term is unknown.

Still, other molecules in myeloid cells might also regulate the apoptotic pathways. For example, we found upregulation of CD5L, also known as apoptosis inhibitor of macrophage (AIM). CD5L can support the survival of macrophages when the cells are challenged with infections or other dangers [109]. Although anti-apoptosis [110] is the most well-recognized function of this molecule, little is known about the intracellular mechanisms underlying CD5L regulation

of apoptosis. Whether this molecule could be modulated during the SIV/HIV to modulate the virus reservoir in macrophages or microglia needs further investigation.

Limitations to our study include the number of animals examined, the study of a single time point during the acute infection period, the lower sensitivity of scRNA-seq, the lack of spatial assessment within the brain of where these populations of cells are present, and the potential differences between SIV infection of rhesus monkeys and HIV infection of people. However, studies of the brain in people are largely limited to post-mortem studies and lack control over conditions. Yet many considerations limit the number of nonhuman primates used for terminal experimental studies. The deposition of sequence data and metadata in publicly accessible databases from our studies and others enables the building of larger analyses with more subjects. These data can be useful in meta-analyses across models and disease states [111]. The ability to purify microglia and macrophages from the brains and examine thousands of cells from each animal enables the identification and study of less prevalent populations. The continued development of spatial transcriptomics, sequencing methods, and other technological and analytic advances will also help alleviate these limitations.

In conclusion, we identified six microglial and two macrophage clusters by performing scRNA-seq to assess the brain myeloid cells in rhesus macaques. In response to the acute SIV infection of a small proportion of cells, all myeloid clusters upregulated the genes related to MHC class I molecules and IFN signaling, which also served as the key connections for other cellular responses to the HIV/SIV infection. The activated microglial and macrophage clusters with more upregulated inflammatory cytokines increased their proportions, and the homeostatic or immunosuppressive myeloid clusters decreased their proportions during acute SIV infection. Among the activated clusters, both a microglial and a macrophage cluster exhibited dysregulation of genes associated with pathways linked to neurodegenerative disorders. Changes in the microglial clusters may contribute to worsening neurological health due to their enhanced ability to produce inflammatory molecules and involvement in cellular senescence.

## Materials and methods

### Ethics statement

Macaques were housed in compliance with the Animal Welfare Act and the Guide for the Care and Use of Laboratory Animals in the NHP facilities of the Department of Comparative Medicine, University of Nebraska Medical Center (UNMC). The primate facility at UNMC has been accredited by the American Association for Accreditation of Laboratory Animal Care International. The UNMC Institutional Animal Care and Use Committee (IACUC) reviewed and approved this study under protocols 19-145-12-FC and 16-073-07-FC.

### Animals

The six male adult rhesus macaques used in this study were purchased from PrimGen (Hines, IL) and New Iberia (LA) and tested negative for the indicated viral pathogens: SIV, SRV, STLV1, Herpes B-virus, and measles; and bacterial pathogens: salmonella, shigella, campylobacter, yersinia, and vibrio. Animals were in a temperature-controlled ($23 \pm 2°$ C) indoor climate with a 12-h light/dark cycle. They were fed Teklad Global 25% protein primate diet (Envigo, Madison, WI) supplemented with fresh fruit, vegetables, and water *ad libitum*. The animal care and veterinary personnel observed the monkeys twice daily for health status. Three of the six animals were intravenously inoculated with a stock of $SIV_{mac}251$ to establish acute SIV infection (93T, 94T, and 95T). The other three macaques were uninfected (92T,

104T, and 111T) and used as control. Virus stocks were provided by the Virus Characterization, Isolation, and Production Core at Tulane National Primate Research Center.

## Viral loads

To determine the viral load in plasma, the blood of infected animals (93T, 94T, and 95T) was collected at 7-day and 12-day post-inoculation of SIV. The EDTA-anticoagulated plasma was separated from blood by centrifugation. Brain and lymphoid organ specimens were taken to determine viral load in tissues. Plasma SIV RNA levels were determined using a gag-targeted quantitative real-time/digital RT-PCR format assay, essentially as previously described, with six replicate reactions analyzed per extracted sample for an assay threshold of 15 SIV RNA copies/ml [112]. As previously described, a quantitative assessment of SIV DNA and RNA in tissues was performed using gag-targeted nested quantitative hybrid real-time/digital RT-PCR and PCR assays [112,113]. SIV RNA or DNA copy numbers were normalized based on quantitation of a single copy rhesus genomic DNA sequence from the CCR5 locus from the same specimen to allow normalization of SIV RNA or DNA copy numbers per $10^6$ diploid genome cell equivalents, as described [114]. Ten replicate reactions were performed with aliquots of extracted DNA or RNA from each sample, with two additional spiked internal control reactions performed with each sample to assess potential reaction inhibition. The viral load in plasma, lymphatic tissues, and brain are shown in S1 Fig and S1 Table.

## Immunohistochemical staining and quantification

Brains were fixed in 10% neutral buffered formalin, embedded in paraffin, cut into 5 μm sections, and mounted on glass slides. For immunohistochemistry, sections were rehydrated, and endogenous peroxidase activity was blocked by a 3% hydrogen peroxide treatment in absolute methanol. A heat treatment with 0.1 M citrate pH 6.39 was performed for antigen exposure. Sections were blocked with 5% Normal Horse Serum (Vector Labs, Burlingame, CA, USA) in PBS and incubated with the primary antibody diluted in the same buffer. Antibodies were targeted against S100A9 Cat# PA5-79949 (Thermo Fisher Scientific, Rockford, IL, USA) at a 1–-10,000 dilution. Biotinylated secondary antibodies (horse anti-rabbit IgG Cat# MP-7401, Vector Labs, Burlingame, CA, USA) were used. Visualization was achieved using Nova Red (Vector Labs, Burlingame, CA, USA). Counterstaining was done using Gill 2 Hematoxylin (StatLab, McKinney, TX, USA).

   The Aperio CS2 scanning system (Leica, Wetzlar, Germany) was used to scan stained brain slides. Each image file was uploaded to the QuPath software [115] to count stained cells. Within the program, a grid of 500 x 500 μm tiles was set on each image. Stained cells were counted in each tile within a defined tissue area, excluding regions near surface of the brain and the Virchow-Robin spaces to remove the associated pia mater and perivascular space. Each tile with an identified stained cell was uploaded into the CellProfiler [116] software. Images were turned to greyscale using the "ColorToGrey" pipeline, with the relative weight of the blue channel set to 30 while the red and green channels were set to 0. Then, the greyscale images were used to input to the "IdentifyPrimaryObjects" pipeline with an upper bound intensity threshold of 0.4, identifying whether the red stained cells were below the threshold intensity to be considered true positive cells. Once confirmed that the cell was positive, it was circled in QuPath. The total number of cells within the defined area was divided by the area measured to calculate the cell density. Photomicrographs were taken on a Leica Diaplan microscope (Wetzlar, Germany) using the SPOT Idea camera (Sterling Heights, MI, USA).

## Isolation of myeloid cells in the brain

Twelve days after viral inoculation, a necropsy was performed on deeply anesthetized (ketamine plus xylazine) animals, following intracardial perfusion with sterile PBS containing 1 U/ml heparin. Brains were harvested, and approximately half of the brain was taken for microglia/macrophage isolation. Microglia/macrophage-enriched cellular isolation was performed using our previously described procedure [117]. Briefly, the brain was minced and homogenized in cold Hank's Balanced Salt Solution (HBSS, Invitrogen, Carlsbad, CA). After being centrifuged, the brain tissue was digested at 37˚ C in HBSS containing 28 U/ml DNase I and 8 U/ml papain for 30 minutes. After digestion, the enzymes were inactivated by the addition of 2.5% FBS, and the cells were centrifuged and resuspended in cold HBSS. The cell suspension was mixed with 90% Percoll (GE HealthCare, Pittsburg, PA) to yield a final concentration of 20% Percoll and centrifuged at 4˚ C for 15 minutes at 550 x g. The microglia/macrophage pallet at the bottom was resuspended in HBSS and passed through a 40 μm strainer to remove cell clumps and/or aggregates. Cells were again pelleted by centrifugation and resuspended in RBC lysis buffer for 3 minutes to eliminate contaminating red blood cells. A final wash was performed before the resulting cells were quantified on a hemocytometer and Coulter Counter Z1. The isolated cells were then sorted for scRNA-seq. The cells in samples 92T, 93T, 94T, and 95T were sorted and sequenced right after the isolation, but the cells in samples 104T and 111T were cryopreserved before the cell sorting and scRNA-seq. The methods for cryopreservation followed our previous study, which was found to maintain the vast majority of the transcriptomic features of fresh isolated microglia/macrophages [117]. Specifically, after isolation as above, the cells were centrifuged at 4˚ C at 550 x g for 5 minutes, and the supernatant was removed. The pellet was dissociated by tapping and then resuspended by the dropwise addition of a solution of 4˚ C 10% DMSO in FBS at a concentration of $10^6$ cells per milliliter. Cells were transferred to cryopreservation tubes and slowly controlled freezing at -80˚ C. After 24 hours, cryotubes were transferred to liquid nitrogen for long-term storage.

## Single-cell preparation and RNA sequencing

For fresh brain isolates, cells were washed in PBS and stained with UV-blue live/dead. Cells were washed, resuspended in MACS buffer with 0.1% BSA, and counted. The cells were stained with non-human primate CD11b microbeads and CD45 microbeads (Miltenyi, San Diego, CA, USA). Four hundred million cells were reconstituted in 320 μL of MACS buffer and reacted with 180 μL of CD11b and 40 μL of CD45 microbeads at 4˚ C for 15 minutes. After incubation, cells were washed, resuspended in 1 ml of MACS buffer, and loaded on MACS Separator LS columns. The double enriched fractions were collected and counted, and then stained with antibody cocktails including BV711-labeled anti-CD20 antibody, BV421-labeled anti-CD3 antibody, BV605-labeled anti-CD11b antibody and PE-labeled anti-CD45 antibody (Biolegend, San Diego, CA) for 45 minutes at 4˚ C. Cells were washed with e-bioscience flow cytometry staining buffer and sorted on Aria2 flow cytometer (BD Biosciences, San Jose, CA, USA). The selection of cells was based on the size, singlet, live, and expression of CD20, CD3, CD45, and CD11b. The CD20 positive cells were all excluded, and the CD20 negative cells that were positive for either CD45 or CD11b or both CD45 and CD11b were collected for scRNA-seq library preparations.

Samples of cryopreserved cell isolates, stored in liquid nitrogen described above, were rapidly thawed in a 37˚ C water bath. The cell recovery procedures were well described in our previous publications [117]. After the recovery, cells were washed and counted by Coulter Counter Z1. Once cell concentration was known, cells were transferred to ice-cold PBS, following the CD45 and CD11b enrichment procedures and FACS sorting procedures.

Post-sorting, isolates were concentrated to approximately 1000 cells per μL and assessed by trypan blue for viability and concentration. Based on 10× Genomics parameters targeting 8000 cells, the ideal volume of cells was loaded onto the 10× Genomics (Pleasanton, CA, USA) Chromium GEM Chip and placed into the Chromium Controller for cell capturing and library preparation. This occurs through microfluidics and combining with Single Cell 3' Gel Beads containing unique barcoded primers with a unique molecular identifier (UMI), followed by lysis of cells and barcoded reverse transcription of RNA, amplification of barcoded cDNA, fragmentation of cDNA to 200 bp, 5' adapter attachment, and sample indexing as the manufacturer instructed with version 3 reagent kits. The prepared libraries were sequenced using Illumina (San Diego, CA, USA) Nextseq550 and Novaseq6000 sequencers. The sequences have been deposited in NCBI GEO (accession number GSE253835).

## Bioinformatics

Sequenced samples were processed using the 10× Genomics Cell Ranger pipelines (7.1.0). Specifically, the scRNA data was demultiplexed and aligned to the customized Mmul10 rhesus macaque reference genome (NCBI RefSeq assembly) and a chromosome representing the SIV genome. For this, overlapping PCR products derived from reverse-transcribed RNA isolated from PBMCs of infected animals 93T and 94T were sequenced using Sanger chemistry and sequences combined to yield a consensus SIV genome. This sequence was deposited in NCBI GenBank, accession number PP236443. After filtering as well as counting the UMI and cell barcode by the Cell Ranger count pipeline, the sequenced samples from the same animals were aggregated together to generate a single file containing feature barcode matrices for the downstream analyses. The counting summary statistics generated by 10x Genomics for each sample are shown in S6 Table. The downstream analyses were then implemented with R (version 4.3).

## Cell clustering and differentially expressed genes (DEGs)

To cluster the cell and find DEGs for each cell cluster, the feature barcode matrices were analyzed with Seurat R package [118] (version: 4.3.0). We removed sparsely expressed genes and low-quality cells and kept genes that had expression in at least 10 cells and the cells with UMI count from 400 to 20,000, gene count from 400 to 10,000, and mitochondrial percentage less than 15%. The final cell counts are shown in S6 Table. Then, the scRNA-seq datasets from six animals were merged to generate a single Seurat Object for further analyses. The normalization, scaling and finding variable genes were performed by SCTransform v2 (SCT) [119]. After normalization, we performed principal component analyses (PCAs) with a default setting of 50 principal components (PCs) to reduce dimensionality. To minimize the batch effect and integrate the datasets, we implemented the harmony [120] (version: 1.0) before clustering. The integrated dataset was then subjected to graph-based clustering, which used the first 30 PCs and 0.2 as resolution. We selected the top 30 PCs which explain approximately 80% of the total variation. The Uniform Manifold Approximation and Projection (UMAP) was used as a nonlinear dimensional reduction method to visualize the cell clusters further. The settings for running UMAP followed the default except for defining the dimensionalities using the first 30 batch effect corrected PCs.

We then characterized each cluster in two ways. First, we examined the expression of general cell markers for microglia (P2RY12, GPR34, and CX3CR1), CNS-associated macrophages (MHC class II), T/NK cells (CD3D, GZMB, and NKG7), B cells (EBF1 and MS4A1), and endothelial cells (RGS5, CLDN5, and ATP1A2) in each cluster. Next, we found the DEGs for each cell cluster by performing the Wilcoxon rank sum test embedded in the FindAllMarkers function of Seurat to the data normalized by SCT. The positive markers with 0.25-fold change (log-

scale) on average detected in a minimum of 25% of cells in either of the two populations were calculated as biomarkers for each cluster (S2 Table). The DEGs found for the same macrophage/microglia cluster but under different infection conditions were also identified using the above test method and criteria (S4 Table). When we used the same way to find DEGs between infected cells and uninfected cells (S2 Fig and S3 Table), to avoid losing the signal for SIV, we performed the FindAllMarkers function on log-normalized data, which was obtained by applying the LogNormalize function in Seurat with scale factor as 10000. The average gene expression for the macrophages/microglia in the same cluster was calculated separately for the uninfected and acute-infected cells. Then, the values were added to 1 and converted to $\log_{10}$ values for plotting. Given our goal in this study to analyze the brain's myeloid cells, we further subsetted the microglial and macrophage clusters and reran the FindAllMarkers function with the aforementioned settings within them separately. The DEGs found in subclusters of microglia or macrophages were further used in Gene Ontology (GO) analyses for further characterization.

## Gene Ontology (GO) and Kyoto Encyclopedia of Genes and Genomes (KEGG) Over-representation analyses (ORA)

Over-representation analysis (ORA) is a widely used approach to determine whether known biological functions or processes are enriched in an experimentally derived gene list (e.g., DEGs), and the p-value in this analysis is calculated by hypergeometric distribution [121]. The GO-ORA and KEGG-ORA were all implemented with the clusterProfiler R package [122] (version: 4.0.2). The GO analysis uses Entrez Gene identifiers instead of the common gene symbol. Therefore, we converted the common gene symbol to their Entrez Gene identifiers by genome-wide annotation package for rhesus macaque (version: 3.18). This package mapped the gene symbol to Entrez Gene identifiers based on NCBI databases (updated Sep-11, 2023). The featured pathways of each microglial or macrophage cluster were then identified using GO-ORA based on the DEGs, and the biological process (BP) was chosen as subontology for analysis. The KEGG-ORA used the DEGs found between different conditions as input to find the upregulated pathways for microglial or macrophage clusters in response to acute SIV infection. The gene annotation of rhesus macaque for KEGG analyses was found in the KEGG database (Mmul10, RefSeq). For both GO-ORA and KEGG-ORA, the cutoff for p-value was set to <0.05. The results were visualized by dot plots, bar plots, tree plots, and gene-concept networks, plotted using the Enrichplot package (version 1.22.0). All GO-ORA and KEGG-ORA pathways detected with p-values less than 0.05 were summarized in S5 Table. In the table, the analysis results provided geneRatio and BgRatio, which are the ratio of input genes annotated in a term and all genes annotated in this term, respectively.

## Trajectory analysis

The monocle3 R package [58,123,124] (version: 1.2.7) was used to estimate lineage differentiation within the macrophage clusters. We extracted the macrophage clusters from the merged Seurat Object and further constructed single-cell trajectories. The trajectory graph was inferred and fitted to the cell clusters generated by Seurat. The Macro-8 cluster was defined as the root node based on the prior knowledge [47] for ordering all macrophages in their pseudotime. For visualization, the UMAP embeddings from Seurat Object were used, the nodes and branches were delineated based on the trajectory analysis, and the cells were colored by their pseudotime. To better compare the pseudo time of cells in Macro-6 and Macro-8, they were further illustrated in a box plot using ggplot2 (version: 3.4.4).

## Statistics

Alpha less than 0.05 was considered as a significant difference in all comparisons. The DEGs were found using non-parametric Wilcox rank sum test, and p-values were adjusted based on Bonferroni correction. In ORA analyses, the enrichment p-value is calculated using hypergeometric distribution, and the p-value was adjusted in GO-ORA analysis to compare multiple microglial clusters. For the quantification of IHC results, the t-test was conducted in GraphPad Prism 10 for comparing the uninfected and acute-infected animals in each brain region independently, and p-values corrected for multiple testing using the two-stage step-up procedure of Benjamini, Krieger, and Yekutieli in Prism.

## Supporting information

**S1 Fig. SIV viral load in plasma, lymphatic tissues, and brain.**
(TIF)

**S2 Fig. Venn diagram of brain myeloid cell upregulated DEGs.**
(TIF)

**S3 Fig. The upregulated pathways for each microglial and macrophage cluster during acute SIV infection.**
(TIF)

**S4 Fig. The comparison of the upregulated genes and downregulated genes in acute SIV infection for microglial or macrophage clusters.**
(TIF)

**S1 Table. Infected cells in tissue and myeloid clusters.**
(XLSX)

**S2 Table. Markers for each cluster.**
(XLSX)

**S3 Table. Myeloid cell comparison.**
(XLSX)

**S4 Table. DEG conditions and clusters.**
(XLSX)

**S5 Table. Pathway analysis.**
(XLSX)

**S6 Table. Seq and cell summary.**
(XLSX)

## Acknowledgments

We thank Drs. Shilpa Buch and Siddappa Byrareddy, Mr. Moses Apostol, Ms. Brenda Morsey (deceased), as well as all members of our non-human primate/SIV collaborative team at UNMC. We also want to acknowledge UMNC Genomics Core Facility and Flow Cytometry Research Facility for the excellent technical support of scRNA-seq and FACS used in this study, the University of Nebraska Lincoln (UNL) Computing Center for high-performance supercomputer access, the Quantitative Molecular Diagnostics Core of the AIDS, the Cancer Virus Program of the Frederick National Laboratory for expert assistance with viral load

measurements, and the Virus Characterization, Isolation and Production Core at the Tulane National Primate Research Center for the SIV$_{mac}$251 stock.

## Author Contributions

**Conceptualization:** Howard S. Fox.

**Data curation:** Meng Niu.

**Formal analysis:** Xiaoke Xu, Meng Niu, Andrew J. Trease, Howard S. Fox.

**Funding acquisition:** Howard S. Fox.

**Investigation:** Xiaoke Xu, Benjamin G. Lamberty, Katy Emanuel, Shawn Ramachandran, Andrew J. Trease, Mehnaz Tabassum, Jeffrey D. Lifson, Howard S. Fox.

**Methodology:** Xiaoke Xu, Meng Niu, Shawn Ramachandran.

**Project administration:** Howard S. Fox.

**Supervision:** Meng Niu, Howard S. Fox.

**Writing – original draft:** Xiaoke Xu.

**Writing – review & editing:** Meng Niu, Shawn Ramachandran, Jeffrey D. Lifson, Howard S. Fox.

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
