## [Decision Letter · Decision Letter 0]

13 May 2024

Dear Dr. Fox,

Thank you very much for submitting your manuscript "Microglia and macrophages alterations in the CNS during acute SIV infection: a single-cell analysis in rhesus macaques" for consideration at PLOS Pathogens. As with all papers reviewed by the journal, your manuscript was reviewed by members of the editorial board and by 3 independent reviewers. In light of the reviews (below this email), we would like to invite the resubmission of a significantly-revised version that takes into account the reviewers' comments. All the reviewers were impressed with the scope and significance of the study. However, 2 reviewers asked for substantial revisions to the manuscript, and perhaps one additional experiment, if feasible (verification

We cannot make any decision about publication until we have seen the revised manuscript and your response to the reviewers' comments. Your revised manuscript is also likely to be sent to reviewers for further evaluation.

Sincerely,

Susan R. Ross, PhD

Section Editor

PLOS Pathogens

Susan Ross

Section Editor

PLOS Pathogens

Michael Malim

Editor-in-Chief

PLOS Pathogens

orcid.org/0000-0002-7699-2064

Reviewer's Responses to Questions

**Part I - Summary**

Reviewer #1: In this study, the authors performed scRNA-seq analysis to investigate the impact of SIV infection on myeloid cells in monkey brains, 12 days after the infection. They focused on analysis of 6 microglial and and 2 macrophage clusters identified by UMAP. Bioinformatic analysis was performed to compare the population sizes, DEG, and biological pathways of individual clusters in naive and infected animals. The data provide an overview of the impact of acute SIV infection on these types of myeloid cells. The finding should be useful for generating hypotheses about their pathogenic contribution.

Reviewer #2: The manuscript explore the different populations of microglia and macrophages in the brain of infected macaques with SIV. The identification of the populations infected as well a the profile of each cell types is just outstanding and will help many labs to understand the role of microglia in SIV/HIV infection.

Reviewer #3: The investigators present a study of the gene expression patterns of brain macrophages (CNS-associated macrophages (CAMs) and microglia during the course of SIV infection, at the single-cell analytical level, with extrapolation of results to putative pathways of neuropathogenesis. Sequencing was performed on isolated brain macrophages at day 12 post-infection, representing the acute phase of infection. Among microglia, they identified 6 clusters of sub-types that vary in their state of activation and production of pro-inflammatory molecules and putative disease-associated molecules. Common among all microglial clusters elevated expression of senescence genes, elevated expression of MHC1 molecules and interferon-related genes (IRGs), consistent with some previous microglial studies in various non-human primate models, reflecting anti-viral responses and associated pathogenic changes. Furthermore, at day 12 reductions in the ‘homeostatic’ class and a ‘pre-activation’ class were observed, interpreted as being consistent with a generally activated state of the microglial population as a whole.

They also identified two emerging classes of CAMs with alternately elevated or reduced CD14 and CD16 (CD14low CD16 hi, and CD14hi CD16low) and an apparent change to a pro-inflammatory state in one CAM population, which predominated.

Finally, one microglial cluster and one CAM cluster associated with expression of pathways known to be associated with neurodegeneration.

This study has important strengths. It is important in its identification of sub-types of microglia and macrophages in the CNS that subserve difference responses to infection and potential consequences for neuropathogenic injury. It extends copious previous work of numerous investigators, and it suggests much more complex intra-lineage responses of myeloid cells to acute infection, which could potentially broaden targets and/or neuroprotection strategies for early therapeutic interventipns in HIV infection.

It is well-designed and rigorously executed, and the data are believable. The deposited data will be useful for other investigators. The study provides novel information about myeloid cell responses in early infection, and it will likely promote additional investigations, particularly leading through chronic infection in non-human primates (NHP). An interesting, novel observation is the elevated expression of an anti-apoptotic molecule, CDL%, but not Bcl-2.

There are several weaknesses. The major weakness is the presentation of a large amount of complex data in multiple figures that are almost illegible in their labeling. There is a great deal of detail in the text about multiple changes in gene expression (DEG), which are difficult to follow without clearer, more legible figures. This generally applies to most/all of the figures. The small number of animals studied (3 experimental, 3 control), which is a common limitation in such primate studies; the investigators recognize this and are appropriately careful in their interpretations.

Other weaknesses are noted. The conclusion that expression of genes in microglia and CAMs reflects SIV infection in those cell types (which is estimated at 0.15%) may be only partially correct. Lymhocytes were not examined in this study, and they may harbor virus at a much higher frequency at this early infection point. Could sub-sets of the microglia and CAMs be affected? This should at least be considered and briefly discussed. Finally, the rationale for choosing a day 12 time point should be discussed specifically with respect to the disease state (day, if possible) being modeled in humans (acute vs. chronic).

other more specific points:

1) In the introduction please update the estimated number of PWH living worldwide—data beyond 2016 should be provided.

2) Please indicate the estimated time-frame (days or weeks) used in defining acute infection as ‘before the development of antibodies to HIV’. Is the time frame in humans/HIV the same as that in non-human primates/SIV?

3) Please also comment or even merely speculate on the relative contribution of lymphocytes vs macrophages in seeding the CNS during this early (12 d) infection. Which predominates? Does the NHP model differ from human infection?

4) Figures should be enlarged. Much of the labeling, axes are illegible.

5) The low % of SIV infection in microglia and CAMs (0.15%) at day 12 suggests that perhaps much of the viral load is expressed from lymphocytes. The suggestion that this might reflect the low sensitivity of scRNA-seq and a high false-negative rate is made. Is the inoculating SIV strain macrophage – tropic or strictly lymphocyte-tropic or both?

6) The assignment of APOE and APOC to neurodegeneration pathways relevant to SIV or HIV pathogenesis is at best speculative. What is true for Alzheimer’s disease or several other neurodegenerative diseases might not be true for this model system. Some references specific for SIV and/or HIV neuropathogenesis risk attributable to APOE and APOC should be provided.

**Part II – Major Issues: Key Experiments Required for Acceptance**

Reviewer #1: 1. the observation that SIV infection seems to alter the population size of every microglial and macrophage cluster indicates a global impact of the infection on these cell types. This type of global changes may be of pathogenic significance. Given that microglial in different CNS regions may respond to HIV-1 protein differently (Zheng at el., iScience

. 2021 Feb 12;24(3):102186.), I wonder if these changes occur in all brain regions or restricted to regions more relevant to HAND. Immunostaining of some of the cluster markers in selected brain regions would be informative.

2. micro-3 and macro-8 clusters are with upregulated genes implicated in neurological diseases. This is a potentially important finding. The point would be much stronger if the expression of representative genes in these cells can be confirmed by another approach (e.g., immunostaining or RNAscope).

Reviewer #2: The manuscript is outstanding. The n is small but acceptable for the expensive and clarity of the data presented. There is not significant outlawyers for monkey studies. The supplemental tables are well described and organized for other labs to use the large data sets for future studies.

Reviewer #3: No new experiments are required, but a major revision of the writing and figure presentation is necessary

**Part III – Minor Issues: Editorial and Data Presentation Modifications**

Reviewer #1: it would be informative to include some discussion on the cause of the global changes of all microglial clusters. is it directly caused by the pathogen (SIV)? or pathogen specific?

Reviewer #2: None

Reviewer #3: see above comments

PLOS authors have the option to publish the peer review history of their article (what does this mean?). If published, this will include your full peer review and any attached files.

Reviewer #1: No

Reviewer #2: **Yes: **Eliseo Eugenin

Reviewer #3: No
---

## [Decision Letter · Decision Letter 1]

26 Aug 2024

Dear Dr. Fox,

We are pleased to inform you that your manuscript 'Microglia and macrophages alterations in the CNS during acute SIV infection: a single-cell analysis in rhesus macaques' has been provisionally accepted for publication in PLOS Pathogens.

Best regards,

Susan R. Ross, PhD

Section Editor

PLOS Pathogens

Susan Ross

Section Editor

PLOS Pathogens

Michael Malim

Editor-in-Chief

PLOS Pathogens

orcid.org/0000-0002-7699-2064

Reviewer Comments (if any, and for reference):

Reviewer's Responses to Questions

**Part I - Summary**

Reviewer #1: The authors have addressed my comments, and the paper is suitable for publishing.

Reviewer #3: The major strength is the identification of sub-clusters of brain microglia that respond differently to SIV infection (in a limited number of cells), which likely underlies neuropathogenic processes. It effectively extends knowledge beyond older studies that clearly identified myeloid lineages as being important for CNS pathogenesis.

**Part II – Major Issues: Key Experiments Required for Acceptance**

Reviewer #1: (No Response)

Reviewer #3: none

**Part III – Minor Issues: Editorial and Data Presentation Modifications**

Reviewer #1: (No Response)

Reviewer #3: none

PLOS authors have the option to publish the peer review history of their article (what does this mean?). If published, this will include your full peer review and any attached files.

Reviewer #1: No

Reviewer #3: No

---

## [Editor Report · Acceptance letter]

11 Sep 2024

Dear Dr. Fox,

We are delighted to inform you that your manuscript, "Microglia and macrophages alterations in the CNS during acute SIV infection: a single-cell analysis in rhesus macaques," has been formally accepted for publication in PLOS Pathogens.

Best regards,

Michael Malim

Editor-in-Chief

PLOS Pathogens

orcid.org/0000-0002-7699-2064